# Structural insights into insect-selective sodium channel toxins drive AI-enhanced biopesticide design

Heng Jiang [1,12], Ruibo Gao[2,3,4,12], Huiqin Xu[5,12], Cheng Wang[6,12], Shuyue Ma[1], Yishu Gong [1], Lianyun Lin[1], Lina Yang[1], Xiang Li[1], Ye Liu[1], Rongcai Lu[2,3,4], Jun-An Ma [7], Jinbo Xu [6], Ke Dong [8], Filip Van Petegem [9], Zheng Liu [5] ✉, Shaoying Wu [2,3,4] ✉ & Zhiguang Yuchi [1,10,11] ✉

Many voltage-gated sodium channel-targeting animal peptide toxins are renowned for their potency and selectivity against insects. Understanding why these toxins selectively target insect sodium channels over their mammalian counterparts is crucial for developing safer and more effective pest control agents. Here, we present the cryoelectron microscopy (cryo-EM) structures of the insect sodium channel $Na_vPaS$ bound to two naturally occurring insect-selective toxins, Av3 from the sea anemone and LqhαIT from the scorpion. Both toxins bind to the voltage-sensing domain 4 (VSD4) of $Na_vPaS$ and disrupt fast inactivation by stabilizing the S4 segment in a deactivated conformation. While Av3 engages a membrane-embedded site between VSD4 and pore domain 1 (PD1), LqhαIT binds to the classical neurotoxin site 3, illustrating distinct binding modes that converge on a shared mechanism of action. These structures reveal the molecular determinants of insect selectivity and highlight the molecular coevolution of toxin-channel interactions, as corroborated by electrophysiology and toxicity assays. Leveraging these insights, we apply AI-driven protein design tools to increase the insecticidal potency of LqhαIT, resulting in a variant with a remarkable doubling in efficacy, as we confirm by insecticidal bioassays. This study illuminates the diverse mechanisms of sodium channel modulation and provides a framework for the structure-guided, AI-driven design of toxin-based biopesticides.

Voltage-gated sodium ($Na_v$) channels, integral membrane proteins found in the plasma membranes of excitable cells, are critical for initiating and propagating action potentials[1,2]. These channels respond to membrane depolarization by rapidly opening, allowing an influx of $Na^+$ ions, and then swiftly transitioning to an inactivated state within milliseconds[3]. Eukaryotic $Na_v$ channels comprise a pore-forming α subunit and auxiliary β subunits[4]. The α subunit contains four homologous domains (DI-DIV), each containing six transmembrane helices (S1–S6). The S1–S4 helices form the voltage-sensing domain (VSD), whereas the S5 and S6 helices, together with two intervening pore

helices (P1 and P2), constitute the pore domain (PD)[5]. Sodium ion selectivity is conferred by the selectivity filter (SF), which is formed by specific residues (D, E, K, and A)[6]. Within the VSD, positively charged residues in the S4 helix are critical for voltage sensing[7]. Upon depolarization, conformational shifts in VSD1-3 open the central pore, whereas the movement of VSD4 triggers fast inactivation[8]. This occurs through binding of the IFM (Ile-Phe-Met) motif, located in the DIII-IV linker, to a cavity outside the S6 bundle, allosterically blocking ion flow[9,10]. Inactivation is also regulated by the C-terminal region, specifically through interactions between an EF-hand domain with the DIII-

IV linker and fibroblast-homologous factors, and by calmodulin binding to an IQ domain[11-13]. This intricate mechanism highlights the structural and functional complexity of Na$_v$ channels, which are finely tuned for precise neural and muscular signaling[14].

Insect Na$_v$ channels are well-established targets for insecticides, including widely used pesticides such as dichlorodiphenyltrichloroethane (DDT), pyrethroids, and indoxacarb[15-17]. However, the extensive use of these chemical pesticides has led to major challenges, including the emergence of resistance in pest populations and environmental concerns. To address these issues, animal toxins found in the venoms of numerous metazoans offer promising alternatives for pesticide development. A notable example is SPEAR® biopesticides developed by Vestaron Corp., which utilize the peptide toxin GS-Omega/Kappa-Hxtx-Hv1a derived from the funnel-web spider *Hadronyche versuta*. When combined with *Bacillus thuringiensis kurstaki* (Btk), a biopesticide that disrupts the insect midgut, orally administered GS-Omega/Kappa-Hxtx-Hv1a can reach its target, nicotinic acetylcholine receptors (nAChRs), in the insect nervous system, where it induces hyperexcitability by enhancing channel opening[18]. This innovative strategy highlights the potential of animal toxins in the development of next-generation green biopesticides that are both effective and environmentally sustainable.

In nature, animal venoms are complex mixtures containing multiple components that target ion channels from both insect and mammalian systems, serving functions related to prey capture and predator defense, respectively. Many animal toxins have evolved to selectively target insect Na$_v$ channels, either by modifying channel gating or through direct channel blocking[19]. Toxins from scorpions, sea anemones, spiders, and cone snails exhibit remarkable selectivity for insect Na$_v$ channels, making them ideal candidates for ecofriendly bioinsecticides. Despite their potential, the molecular mechanisms underlying their insect-specific selectivity remain poorly understood. Unraveling how these toxins interact with and modulate Na$_v$ channels not only advances our understanding of channel function but also opens the door to the design of highly specific and sustainable bioinsecticides[20,21].

Av3 and LqhαIT are two cysteine-rich peptide toxins with remarkable insect selectivity. Av3, a 27-amino-acid toxin derived from the venom of the sea anemone *Anemonia viridis*[22], and LqhαIT, a 66-amino-acid α-scorpion toxin (ScTx) found in the venom of the deathstalker scorpion *Leiurus hebraeus*, specifically target Na$_v$ channels in arthropods such as insects and crustaceans[23]. Importantly, both toxins exhibit minimal effects on mammalian Na$_v$ channels, positioning them as promising candidates for developing eco-friendly bioinsecticides[24-26]. The structures of these toxins are stabilized by multiple disulfide bonds, three in Av3 and four in LqhαIT. Structurally, Av3 adopts a compact, turn-based conformation, whereas LqhαIT features the characteristic β1-α1-β2-β3 scaffold shared by all ScTxs[22,27,28]. Previous mutagenesis studies suggest that these two toxins share a common but not identical binding site near the extracellular region of VSD4, referred to as neurotoxin binding site 3[20,25,28]. By targeting this site, they inhibit fast inactivation of the Na$_v$ channel, leading to prolonged neuronal firing, paralysis of insect muscles, and ultimately death. Despite extensive research, the precise receptor sites and molecular mechanisms underlying their modulation and insect-specific selectivity remain incompletely understood.

Advances in cryo-EM have provided significant insights into the structural basis of Na$_v$ channel modulation by animal toxins[29-35]. For example, structural studies of human Na$_v$1.7 with the mammal-selective spider toxins HWTX-IV and ProTx-II have revealed their distinct binding sites and provided insights into their mechanisms of Na$_v$ channel gating modulation. However, structural data on Na$_v$ channels bound to insect-selective toxins are largely unavailable. This knowledge gap limits our understanding of species-specific selectivity and toxin-channel coevolution, as well as the development of insect-selective biopesticides. Further structural studies are critical to unravel

the molecular basis of insect selectivity and to advance the design and optimization of toxin-based biopesticides.

In this study, we resolve the cryo-EM structures of the American cockroach Na$_v$ channel Na$_v$PaS in complex with the insect-selective toxins Av3 and LqhαIT at resolutions of 2.8 Å and 2.9 Å, respectively. Using mutagenesis, electrophysiology, and surface plasmon resonance (SPR), we identify key residues responsible for species-specific recognition and reveal toxin-induced conformational changes that disrupt fast inactivation. Additionally, by leveraging complex structure analysis and artificial intelligence (AI)-driven design, we engineer a mutant LqhαIT variant with doubled potency against insect Na$_v$ channels. This work elucidates the molecular mechanisms underlying insect selectivity and offers structural templates for the development of biopesticides.

## Results

### Av3 and LqhαIT selectively modulate insect Na$_v$ channels

Av3 and LqhαIT are natural toxins produced by sea anemones and scorpions to prey on arthropods. To assess their species-specific modulatory properties, we compared their activities on insect and mammalian Na$_v$ channels. Na$_v$PaS, a Na$_v$ channel from the American cockroach *Periplaneta americana*, is highly stable and retains essential Na$_v$ channel features, making it ideal for structural studies. However, it does not conduct sodium currents when expressed in *Xenopus* oocytes. Therefore, for electrophysiological analyses, we employed the orthologous insect Na$_v$ channel, BgNa$_v$1-1a, from the German cockroach *Blattella germanica*, which was also used in a previous study to functionally validate the Na$_v$PaS-Dc1a complex structure[29]. BgNa$_v$1-1a shares ~59% sequence identity with Na$_v$PaS, with higher conservation in the extracellular toxin-interacting regions, making it a suitable surrogate for electrophysiological studies[36,37]. As a control, we used the cardiac isoform of the human Na$_v$ channel (hNa$_v$1.5), which provides preliminary assurance of mammalian safety. hNa$_v$1.5 is representative of mammalian Na$_v$ channels and shares 42% overall sequence identity with Na$_v$PaS. Both Av3 and LqhαIT selectively modulate BgNa$_v$1-1a by significantly increasing the peak Na$^+$ current and inhibiting fast inactivation at nanomolar concentrations, with EC$_{50}$ values of 191.6 nM and 321.9 nM, respectively (Fig. 1A, B). In contrast, both toxins had minimal effects on hNa$_v$1.5, even at the highest tested concentration of 40 μM (Fig. 1A, B), highlighting their strong insect selectivity.

### Av3 and LqhαIT target distinct sites on VSD4

To investigate the molecular mechanisms by which Av3 and LqhαIT modulate insect Na$_v$ channels, we determined the cryo-EM structures of Na$_v$PaS bound to these toxins at resolutions of 2.8 Å and 2.9 Å, respectively. Previous studies have shown that Av3 competes with other site 3 toxins, suggesting that it targets the same or overlapping regions within VSD4[38]. This was further supported by docking studies showing that Av3 sits on top of VSD4[39]. Mutagenesis experiments have also indicated that certain mutations within site 3 differentially affect Av3 and scorpion toxins, implying distinct binding modes, although the precise binding pose of Av3 remains unclear[25]. Our cryo-EM structure of Na$_v$PaS-Av3 reveals that Av3 occupies a membrane-embedded site on Na$_v$ channels. Owing to its small size, high flexibility, and hydrophobic nature, most part of Av3 partitions into the membrane and is sequestered within a deep transmembrane cleft formed between VSD4 and the S5/S6 helices of PD1. The C-terminal tail of Av3 extends beyond the membrane to interact with the extracellular glycan moiety of PD1, increasing the stability of its binding (Fig. 1C).

In contrast, LqhαIT binds to the extracellular side of VSD4, interacting with the glycan moiety from PD1 and positioning itself above the membrane (Fig. 1D). This binding site closely resembles the region targeted by other known α-ScTxs, including AaH2, Lqh3, and MTα-5[30,33,40] (Supplementary Fig. 1). Notably, the binding sites of Av3 and LqhαIT are adjacent with minimal overlap, underscoring their

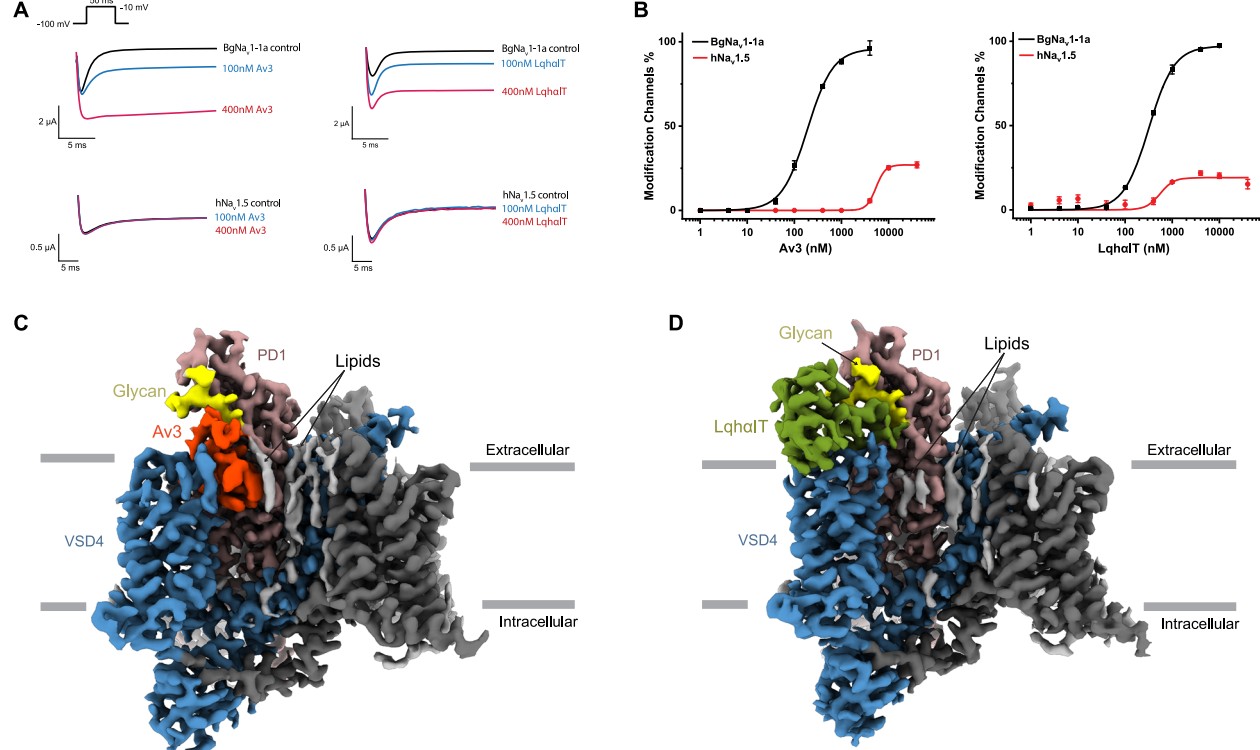

**Fig. 1 | Modulation of Na$_v$ channels by Av3 and LqhαIT. A** Av3 and LqhαIT significantly inhibit fast inactivation of the insect Na$_v$ channel BgNa$_v$1-1a, while showing negligible effects on the mammalian Na$_v$ channel hNa$_v$1.5. Representative sodium current traces are shown at toxin concentrations of 0 nM (black), 100 nM (blue) and 400 nM (red), respectively. **B** Dose-dependent curves showing selective divergent modes of interaction with the channel (Supplementary Fig. 2A, B).

modulation of BgNa$_v$1-1a by Av3 and LqhαIT, with EC$_{50}$ values of 191.6 ± 20.7 nM and 321.9 ± 6.6 nM, respectively, while having minimal effects on hNa$_v$1.5. Data are shown as mean ± SEM ($n = 3$ cells). Source data are provided as a Source Data file. **C, D** Cryo-EM maps of the Na$_v$PaS-Av3 (**C**) and Na$_v$PaS-LqhαIT (**D**) complex structures, highlighting the binding sites on the channel.

## Av3 and LqhαIT bindings induce conformational changes in Na$_v$PaS

Although Av3 and LqhαIT interact with Na$_v$PaS through distinct mechanisms, both toxins induce a similar conformational change in the VSD4. In the unbound state, VSD4 adopts a partially activated conformation, as observed in apo-Na$_v$PaS cryo-EM structure (PDB ID: 5X0M). In this state, two gating charges, R2 and R3, are positioned above the hydrophobic constriction site (HCS) (Phe1213) and interact with the extracellular negative cluster (ENC), which includes Asn1183, Asp1190 and Asn1206. Upon Av3 or LqhαIT binding, the S4 gating charges shift one position down along the gating charge-transfer pathway, converting the VSD from a partially activated state to a deactivated state. In this toxin-bound state, R2 remains coordinated to Asn1183 and Asn1206, while R3 aligns with the HCS and R4 undergoes a significant ~10 Å downward shift across the HCS to interact with the intracellular negative cluster (INC) (Fig. 2A, Supplementary Movie 1).

This downward movement of the S4 helix subsequently generates two electrostatic interfaces: the one between Arg1277 (R6) from the S4 and Arg1293 from the S4-S5 linker with the acidic residues on the C-terminal EF-hand domain (Interface I), and the other between Asp1420 from the S6 and the basic residues on the DIII-IV linker (Interface II). In the apo-Na$_v$PaS structure, only one electrostatic bridge is observed (Asp1420 with Arg1138 from Interface II). Av3 induces the formation of three additional electrostatic bridges, including Arg1277 (R6)-Glu1435, Arg1293-Asp1427, and Asp1420-Arg1142. In contrast, LqhαIT binding induces only two of these bridges, including Arg1277 (R6)-Glu1435 and Asp1420-Arg1142, alongside the pre-existing Asp1420-Arg1138 interaction (Fig. 2B). These interactions contribute

to restraining the DIII-IV linker, thereby impairing the fast inactivation of the sodium channel. This mechanism aligns with the structural changes induced by the toxin AaH2, further supporting a common role of electrostatic interactions in toxin-induced channel modulation[30]. However, several key interacting residues in Na$_v$PaS, such as Arg1138 and Arg1293, are substituted by neutral residues (e.g., alanine, methionine, or asparagine) in other Na$_v$ isoforms (Fig. 2C). These differences suggest that the precise electrostatic network and conformational changes induced by toxin binding may not be fully conserved across other Na$_v$ channels.

Furthermore, Av3 binding expands the binding pocket by wedging itself between VSD4 and PD1, simultaneously pushing the glycan and PD1 toward the pore. Despite this structural rearrangement, the pore size remains unchanged, indicating minimal impact on single-channel conductance or ion selectivity (Supplementary Fig. 3). Additionally, Av3 binding induces the formation of a rigid α-helix in the loop connecting helices S3 and S4, suggesting that toxins can alter the secondary structure of ion channels and supporting an induced-fit binding model (Fig. 2D). In contrast, LqhαIT primarily interacts with VSD4 and lightly leans on PD1, affecting the conformation of the S3-S4 loop with minimal impact on PD1 (Fig. 2D).

## Insect-specific binding determinants of Av3

The Av3-binding site spans three regions: VSD4, the S5-S6 helices of PD1, and the glycan moiety of PD1 (Fig. 3A). The most extensive interaction occurs at VSD4, where ~350 Å² of surface area is buried by Av3. This binding is driven primarily by hydrophobic interactions, complemented by specific hydrogen bonds. Key residues at the VSD4 interface, including Pro5, Tyr7, Trp13, and Tyr18, contribute to Av3 activity (Fig. 3C). Electrophysiological assays show that alanine substitutions at these positions reduce Av3 efficacy by 2.1-, 2.6-, 25.3-, and

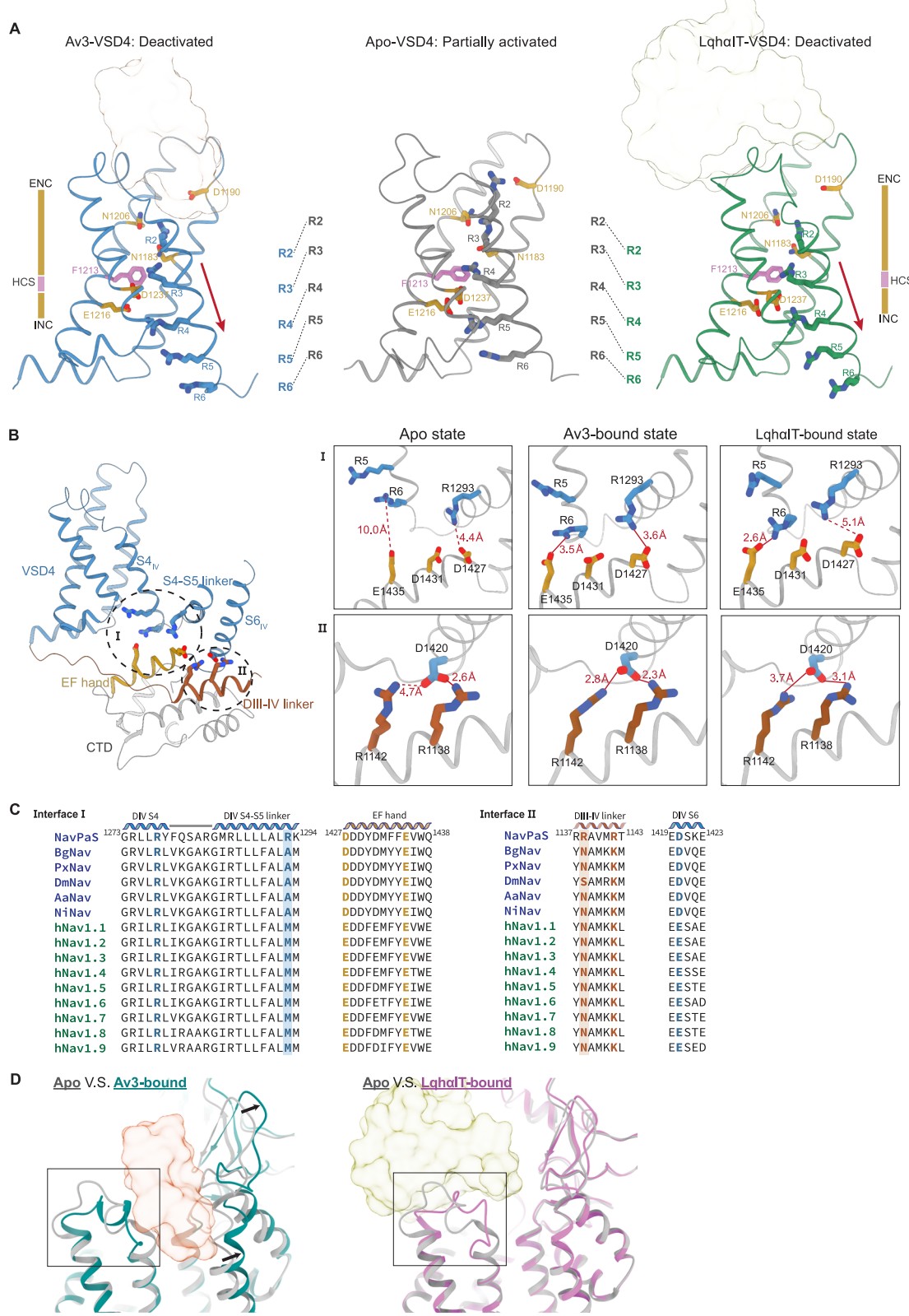

**Fig. 2 | Conformational changes in Na$_v$PaS induced by Av3 and LqhαIT.**
**A** Comparison of the VSD4 conformations in the Av3-bound state (left), apo state (middle) and LqhαIT-bound state (right). Upon toxin binding, VSD4 transitions from a partially activated state to a deactivated state, with gating charges (R2-R6) sequentially moving downward and R4 crossing the hydrophobic constriction site (HCS). **B** Two electrostatic interfaces formed upon toxin binding: (I) between DIV S4 and the S4-S5 linker with the EF hand; and (II) between DIV S6 and the DIII-IV linker. Several salt bridge pairings are altered upon toxin binding. Solid lines indicate salt bridges present in the current state, while dashed lines denote those

formed in an alternate state. **C** Sequence alignment of representative insect and human Na$_v$ channels at the identified electrostatic interfaces. Charged residues involved in salt bridge formation are shown in bold. Shading highlights non-conserved charged residues specific to Na$_v$PaS, which may contribute to isoform-specific interaction networks in this region. **D** Superposition of the Na$_v$PaS apo-state structure with the Na$_v$PaS-Av3 complex (left) and with the Na$_v$PaS-LqhαIT complex (right). Av3 binding induces α-helix formation in the DIV S3-S4 linker and movement of PD1, while LqhαIT binding causes displacement of the DIV S3-S4 linker.

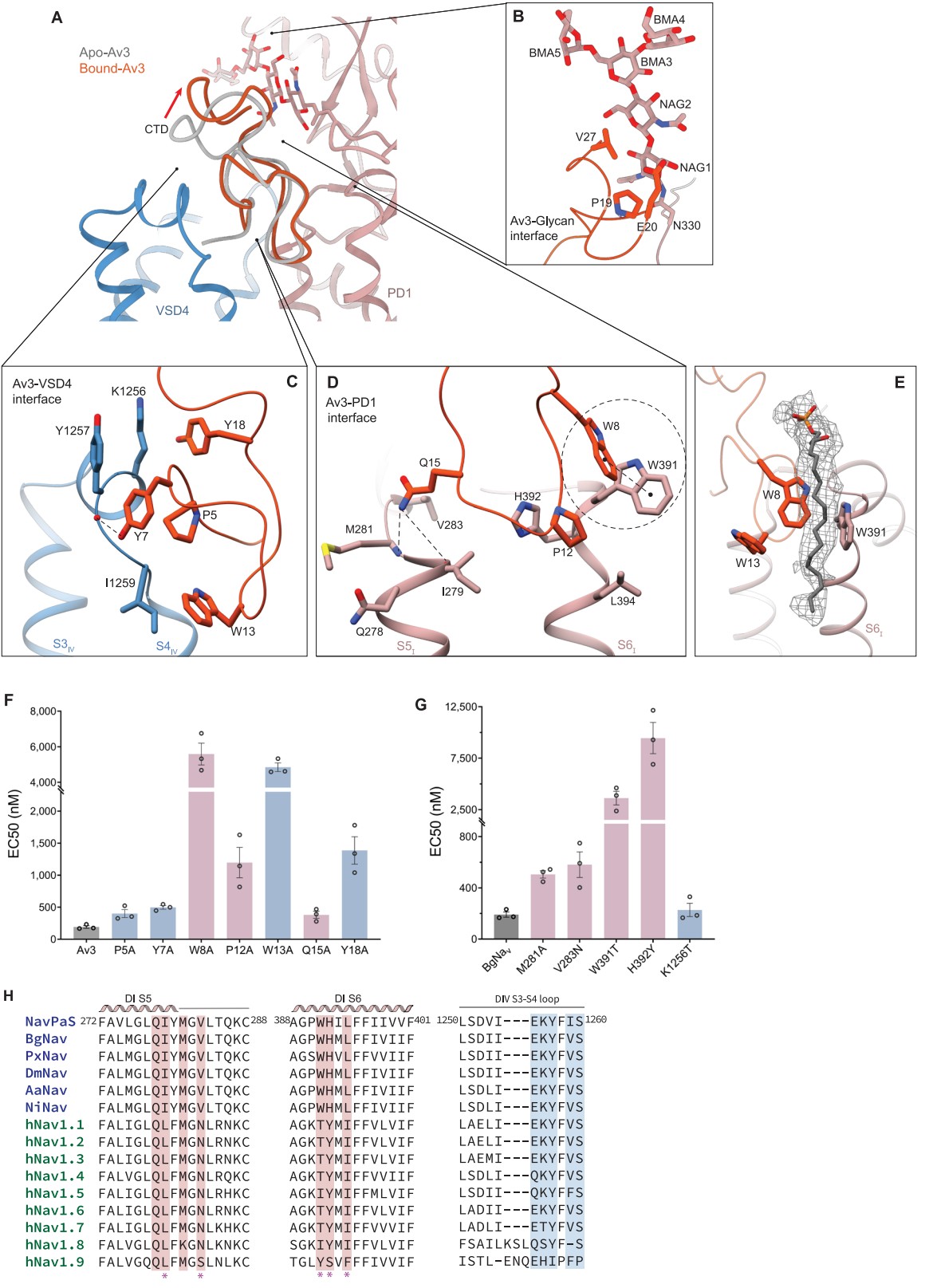

**Fig. 3 | Complex structure of NavPaS-Av3. A** Side view of the Av3-Na$_v$PaS receptor site, with Na$_v$PaS-bound Av3 (orange) superimposed onto its apo state (gray, PDB ID: 1ANS). **B–D** Enlarged views of key interactions at the Av3-glycan (**B**), Av3-VSD4 (**C**), and Av3-PD1 (**D**) interfaces, with hydrogen bonding and Pi-Pi interactions indicated by dashed lines. **E** A putative phospholipid was modeled into a well-defined density (contoured at 5σ) located between Trp8 of Av3 and Trp391 of PD1, suggesting a potential role in mediating the interaction between Av3 and Na$_v$PaS. **F,**

**G** Mutagenesis combined with electrophysiological EC$_{50}$ measurements identifying key residues for Av3 activity on the Av3 side (**F**) and the Na$_v$ side (**G**). Interactions at the PD1 interface are shown in rosy brown, and those at the VSD4 interface in steel blue. All data are shown as mean ± SEM ($n = 3$ cells). Source data are provided as a Source Data file. **H** Sequence alignment of various Na$_v$ channels, focusing on Av3-binding interfaces. Key interacting residues are shaded, and invertebrate-specific residues are marked with an asterisk.

7.2-fold, respectively, highlighting their functional relevance (Fig. 3F). Surface plasmon resonance (SPR) analysis further confirmed a marked loss of binding for the Y7A, W13A, and Y18A mutants (Supplementary Table 1). At the second interface, residues Trp8, Pro12, and Gln15 interact extensively with the S5 and S6 helices of PD1, with Trp8 forming a Pi-Pi interaction with Trp391 (Fig. 3D). Alanine substitutions at these residues notably reduce the binding affinity and efficacy of Av3, with Trp8 and Pro12 having more pronounced effects (Fig. 3F, Supplementary Table 1). The glycan interface, involving Pro19, Glu20, and Val27, also contributes to Av3 binding. However, alanine substitutions at these positions have minimal impact, suggesting a supporting role in the overall interaction[25] (Fig. 3B).

Additionally, some hydrophobic residues in Av3, such as Trp8 and Trp13, face the toxin-lipid interface in our complex structure, suggesting that they may help stabilize the toxin conformation through interactions with the lipid membrane. A well-defined lipid-like density was observed near Trp8 of Av3 and Trp391 of PD1 (Fig. 3E). Based on its shape and position, analogous to lipids observed at equivalent sites in other human Na$_v$ structures, it was modeled as a putative phospholipid, suggesting a conserved lipid-channel interaction[35,41,42]. Notably, Av3 does not displace the lipid but rather leans on it, forming a sandwich-like arrangement with Na$_v$PaS. Electrophysiological assays showed that substitution of Trp8 and Trp13 with alanine markedly reduced Av3 activity (Fig. 3F), supporting a contributing role of membrane interactions in Av3 function.

Binding to Na$_v$PaS induces significant structural changes in Av3 itself. The apo structure of Av3, determined by NMR, has no defined secondary structure, reflecting its high flexibility[22]. This flexibility likely allows Av3 to fit into the narrow niche between PD1 and VSD4. The root mean square deviation (RMSD) between the Cα atoms of the bound and apo forms of Av3 is ~3 Å, with the glycan-interacting C-terminal region exhibiting a more substantial displacement of ~7 Å upon binding (Fig. 3A).

Av3 is highly active against insects and crustaceans but is inactive in mammals. While most interacting residues at the VSD4 interface are conserved across species, our complex structure identified five invertebrate-specific residues in PD1, including Ile279, Val283, Trp391, His392, and Leu394 (Fig. 3D, H). Electrophysiological assays demonstrated that substituting Val283, Trp391, and His392 with their mammalian counterparts significantly reduced the efficacy of Av3 by 3.0-, 16.3- and, 49.4-fold (Fig. 3G), underscoring their importance in species selectivity. Sequence- and structure-based alignment of toxin-binding regions revealed a high degree of conservation across hNa$_v$1.1-1.9, suggesting that Av3's lack of activity against hNa$_v$1.5 likely extends to other human Na$_v$ isoforms (Fig. 3H, Supplementary Fig. 2C, D).

Additionally, the glycan moiety linked to Asn330, which contributes to Av3 binding, is conserved across species. However, variations in glycosylation patterns may influence toxin selectivity (Supplementary Fig. 4), warranting further investigation into the role of these modifications in mediating toxin specificity.

## Insect-specific binding determinants of LqhαIT

The binding of LqhαIT is mediated primarily by interactions involving the β1-α1 loop, β2-β3 loop, and C-terminal tail (CT) of LqhαIT with VSD4, covering a buried surface area of ~450 Å². This is further complemented by interactions between the CT and the glycan of PD1, which contribute an additional buried surface area of ~246 Å² (Fig. 4A–C). Several key contact residues are conserved across all the α-ScTxs. For example, Asn44 from the β2-β3 loop in the VSD4 interface and His64 from the CT in the glycan interface are crucial for binding. Substituting these residues with alanine reduces LqhαIT activity by 47.1-fold and 37.1-fold, respectively (Fig. 4F). On the channel side, Asp1252 from the S3-S4 loop, which interacts with Gly43 from LqhαIT, is a highly conserved residue essential for α-ScTx binding[43]. In line with

its critical role, the D1252A mutation reduces LqhαIT activity 98.2-fold, whereas the D1252R mutation completely abolishes it (Fig. 4C, G).

The overall structure of LqhαIT remains largely unchanged upon binding, with a Cα RMSD of ~0.77 Å compared with the apo structure, indicating that LqhαIT is more rigid than Av3. However, the side chains of several residues near the LqhαIT-VSD4 interface, including Glu15, Phe17, and Arg18, undergo coordinated conformational shifts toward the S3-S4 loop upon binding (Fig. 4D). These movements likely reflect an induced-fit mechanism driven by hydrophobic packing, electrostatic complementarity, and local structural accommodation. Among them, the conserved Arg18 rotates ~120°, but does not form direct contacts with Na$_v$PaS. However, the R18D mutation abolishes toxin activity, suggesting an indirect role in toxin-channel recognition, as confirmed by electrophysiology and SPR (Fig. 4F, Supplementary Table 1). We propose that Arg18 may assist in toxin orientation via electrostatic steering or transient membrane interactions prior to stable binding.

While conserved residues are critical for general toxin-channel recognition, the species specificity of α-ScTxs is primarily dictated by non-conserved surface residues. α-ScTxs can be broadly classified into insect-selective and mammal-selective subtypes, exemplified by LqhαIT and AaH2, respectively. Although they share a conserved βαββ structural scaffold, they diverge at key surface-exposed positions that define their Na$_v$ channel subtype selectivity. Compared with rNa$_v$1.2 and hNa$_v$1.5, LqhαIT displays over 1000-fold selectivity for *Drosophila melanogaster* Na$_v$ (DmNa$_v$), whereas AaH2 demonstrates more than 100-fold selectivity for rNa$_v$1.2 over DmNa$_v$[24,44]. To explore the molecular basis of α-ScTx species selectivity, we compared the binding modes observed in our Na$_v$PaS-LqhαIT structure with the previously published structure of AaH2 bound to the VSD of hNa$_v$1.7 in a Na$_v$1.7/Na$_v$PaS chimeric channel. Our analysis identified two key interfaces contributing to insect specificity. At the VSD4 interface, Lys41 from the β2-β3 loop of insect-selective α-ScTxs is in close proximity to a negatively charged site on the S2 helix, formed by two negatively charged residues, Glu1200 and Asp1203 (Fig. 4E). In contrast, mammal-selective α-toxins feature a proline at this position. Sequence alignment indicates that most mammalian Na$_v$s possess a neutral interface at this site, with the exception of Na$_v$1.4 and Na$_v$1.7, which contain a single negatively charged residue (Fig. 4I). As expected, mutating Lys41 to proline significantly reduces LqhαIT activity against insect Na$_v$ channels. Additionally, LqhαIT residues that face the S3-S4 linker, including Lys8, Val13, and Phe17, play key roles in conferring insect specificity. Substituting these residues with their counterparts from mammal-selective α-ScTxs reduce LqhαIT activity by 45.2-fold, 10.5-fold, and 26.0-fold, respectively (Fig. 4C, F). On the channel side, substituting the insect-specific Ile1253 in the S3-S4 linker with leucine, the corresponding residue in mammalian Na$_v$s, leads to a significant reduction in LqhαIT activity (Fig. 4G, I), highlighting the importance of species-specific interactions in determining toxin sensitivity.

At the PD1 interface, nonconserved residues in the β1-α1 loop and C-terminal region of α-ScTxs, which interact with distal glycan regions, significantly influence toxin selectivity. While the glycans near the conserved glycosylation site are relatively similar across Na$_v$ channels, their distal regions vary (Supplementary Fig. 4). Substituting insect-selective LqhαIT sequences in this region with those from mammal-selective AaH2 greatly reduces activity, including the N9D, Y10V, I57T, and V59G substitutions which significantly decrease LqhαIT activity (Fig. 4B, F, H). Although glycosylation in *Xenopus* oocytes may not reflect the native insect or mammalian patterns, the observed differences suggest that the distal glycan structure may plausibly contribute to differential toxin binding and selectivity.

For selected mutants, binding affinities were also assessed by SPR (Supplementary Table 1), which showed consistent trends with the electrophysiological results. Given the conserved nature of these binding interfaces, the observed insect selectivity of LqhαIT is also

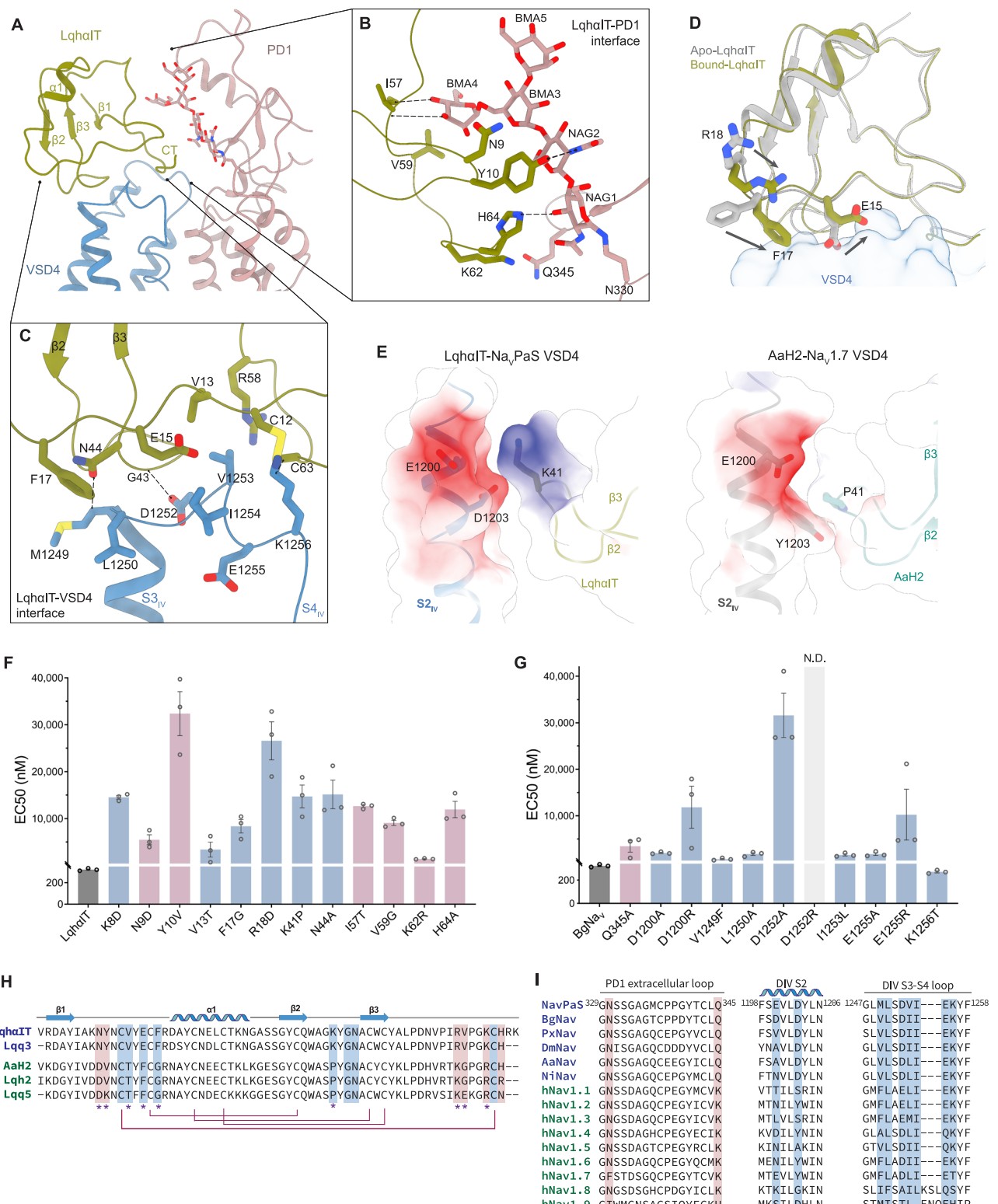

**Fig. 4 | Complex structure of NavPaS-LqhαIT. A** Side view of the LqhαIT-NaᵥPaS receptor site. **B, C** Enlarged views of interactions at the LqhαIT-PD1 (**B**) and LqhαIT-VSD4 (**C**) interfaces, with hydrogen bonding indicated by dashed lines. **D** Comparison of LqhαIT in its apo state (gray, PDB ID: 2ASC) and NaᵥPaS-bound state (olive green), showing structural changes upon binding. **E** Electrostatic surface comparison of the NaᵥPaS-LqhαIT and hNaᵥ1.7-AaH2 (PDB ID: 6NT4) binding interfaces, highlighting that NaᵥPaS-LqhαIT interaction is more driven by electrostatic forces. **F, G** Mutagenesis and electrophysiological EC₅₀ measurements

revealing key residues for LqhαIT activity on the LqhαIT side (**F**) and the Naᵥ side (**G**). PD1 interactions are shown in rosy brown, and VSD4 interactions in steel blue. N.D. activity not detectable. All data are shown as mean ± SEM ($n$ = 3 cells). Source data are provided as a Source Data file. **H, I** Sequence alignment of α-ScTxs (**H**) and Naᵥ channels (**I**), focusing on the LqhαIT-binding interfaces. Key interacting residues are shaded. Insect-selective α-ScTxs are marked in indigo, mammalian-selective α-ScTxs in green, and invertebrate-specific residues with an asterisk.

likely to apply across mammalian Na$_v$ isoforms (Fig. 4I, Supplementary Fig. 2C, D). Together, these findings offer structural insights into the molecular determinants of α-ScTx selectivity and may guide the design of more selective and effective neurotoxins for various applications.

## AI-assisted structure-based design of LqhαIT-derived biopesticides

LqhαIT has demonstrated high potency against a broad spectrum of pest insects while exhibiting low activity on mammalian Na$_v$s, making it an attractive candidate for development as a biopesticide. Leveraging the complex structure of Na$_v$PaS-LqhαIT, we implemented an AI-driven design pipeline that integrates three core models to identify mutations that enhance the toxin's properties (Fig. 5A).

First, we employed our self-developed ComplexDDG algorithm to predict beneficial mutations that could enhance toxin-channel binding affinity, using the complex structure as input. ComplexDDG integrates sequence- and structure-level representations within a geometric deep learning framework (Fig. 5B). Residue embeddings derived from the pretrained protein language model ESM-1v are used as the initial node features of a geometric attention network, capturing evolutionary, structural, and contextual information[45,46]. These are combined with 22 physicochemical energy terms calculated by FoldX, which quantify energetic differences between the wild-type and mutant complexes[47]. The fused sequence-structure-energy features are then used to predict the binding free-energy change (ΔΔG). Based on these predictions, 101 LqhαIT mutations with ΔΔG values below zero were shortlisted as candidates for further analysis.

Next, Evolutionary Scale Modeling 2 (ESM2), a large language model designed to predict the likelihood of amino acid substitutions based on natural mutation probabilities, was applied to identify stability-enhancing mutations[48]. Among the initial candidates, 30 mutations commonly selected by both ComplexDDG and ESM2 were further refined. To ensure the feasibility of protein expression and purification, we used the solubility prediction model Aggrescan3D (A3D) 2.0 to exclude mutations likely to cause aggregation[49], discarding candidates with solubility scores above 1.5. Additionally, we applied manual filtering based on empirical knowledge and included substitutions targeting key sites identified by ComplexDDG. Ultimately, 28 mutations were selected for experimental validation.

The candidate mutants were initially screened via SPR binding assays. Five mutants presented a greater than twofold increase in binding affinity (Fig. 5C). Electrophysiological experiments confirmed functional enhancements, with mutants E15F and A39L showing significant improvements in modifying BgNa$_v$1-1a (Fig. 5D). In vivo activity was assessed by injecting the mutants into *Galleria mellonella* larvae, a representative lepidopteran pest. Both mutants demonstrated enhanced toxicity compared to wild-type LqhαIT, with A39L achieving a remarkable doubling in insecticidal efficacy (Fig. 5E).

Glu15 is located within the β1-α1 loop of LqhαIT and interacts with Val1253 and Ile1254 of the S3-S4 loop. Substituting glutamate with the bulkier, hydrophobic phenylalanine residue likely enhances hydrophobic interactions within this region (Fig. 5F). Interestingly, Ala39, located in the β2-β3 loop, does not directly interact with Na$_v$ but instead faces the aqueous and membrane interface (Fig. 5F). The A39L mutation likely stabilizes interactions with membrane lipids, facilitating toxin docking and insertion into the membrane. This additional pivot point may enhance the toxin binding process. These findings highlight the power of structure-based and AI-driven strategies in advancing the development of highly effective biopesticides.

## Discussion

A major challenge in modern agriculture is the widespread resistance that has developed in many destructive pest species. Most conventional chemical pesticides inevitably face resistance after several years of use. In this context, peptide toxins derived from animal venoms have emerged as a promising reservoir for next-generation biopesticides. These peptides typically bind to ion channels or receptors with large, multivalent interfaces, making it more difficult for single-point mutations to confer resistance. Their distinct mechanisms of action and high species specificity, shaped by hundreds of millions of years of evolution, offer advantages such as minimal off-target effects and no cross-resistance with current insecticide classes. However, the lack of detailed structural information on the target proteins and their interactions with toxins limits our understanding of their binding sites, mechanisms of action, and selectivity. This knowledge gap hinders the rational selection of toxin peptides from large, diverse libraries and the design of more potent, selective, and environmentally friendly biopesticides that are effective in controlling resistant pests.

Recent advances in single-particle cryo-EM have enabled the high-resolution structural elucidation of several Na$_v$ channels in complex with small molecules and peptide ligands, revealing their binding sites and mechanisms. Traditionally, Na$_v$ ligand binding sites are categorized into seven groups based on primary sequences. However, a more recent classification based on structural locations has expanded this classification to 15 sites, including E, S, C, G, I, BIG, F1, F3, F4, V2EP, V2EM, V4EM, V4EC, and V1EM[50]. According to this updated classification, LqhαIT binds to the V4EM site, a region shared with several previously characterized gating modifier toxins (GMTs), such as AaH2 and LqhIII[30,33]. Interestingly, unlike all previously resolved GMTs, which target sites such as V2EP, V2EM, V4EM, and V1EM, the toxin Av3 binds to a distinct GMT site, defined here as V4EP. This distinction sets Av3 apart, as it partially inserts into the extracellular cavity formed by DIV S4, PD1, and the cell membrane, leveraging its natural flexibility and hydrophobicity. Acting as a wedge, Av3 restricts the movement of VSD4, thereby altering channel gating. Multiple sequence alignment of the Av3-binding site (V4EP) revealed high species specificity, underscoring its potential as a bioinsecticide (Fig. 3H). The distinct characteristics of this pocket make it an excellent candidate for developing insect-specific toxins with minimal off-target effects on non-pest species. Additionally, sequence alignment of human Na$_v$s identified several subtype-specific residues within this membrane-involved pocket (Fig. 3H). Given the critical role of this pocket in sodium channel gating modulation, it presents opportunities for the development of subtype-selective therapeutic agents. These findings demonstrate how structural and mechanistic insights can guide the rational design of innovative biopesticides and precision-targeted human therapeutics.

Both Av3 and LqhαIT exhibit high insect selectivity but employ distinct strategies. The insect-selective determinants of Av3 primarily target the PD1 region, whereas the selectivity of LqhαIT arises mainly from its interactions with VSD4. Additionally, while the basal portion of the glycosylation at Asn330 is relatively conserved across Na$_v$ channels, the distal region displays structural heterogeneity among different Na$_v$ isoforms (Supplementary Fig. 4). Notably, the CTD and the β1-α1 loop of LqhαIT, which interact with this distal region, are also conserved among insect-selective α-ScTxs, suggesting a potential role for this glycan surface in modulating toxin-channel recognition (Fig. 4H). These structural features in insect Na$_v$s present opportunities for the future design of biopesticides. Exploiting these distinctions could further enhance the insect selectivity of peptide toxins, minimizing off-target effects and increasing their effectiveness.

Our study highlights the potential of AI to substantially accelerate the design and optimization of toxin-based biopesticides. By combining structure-guided modeling with machine learning-driven prediction tools, we efficiently identified key interface residues and prioritized mutations that enhance toxin-channel interactions. Notably, several AI-predicted mutations in LqhαIT led to measurable improvements in binding affinity and insecticidal activity, as confirmed by SPR and electrophysiological assays. In addition, the expressibility and structural stability of these variants were further optimized using two complementary AI tools, ESM2 and A3D, which facilitated candidate

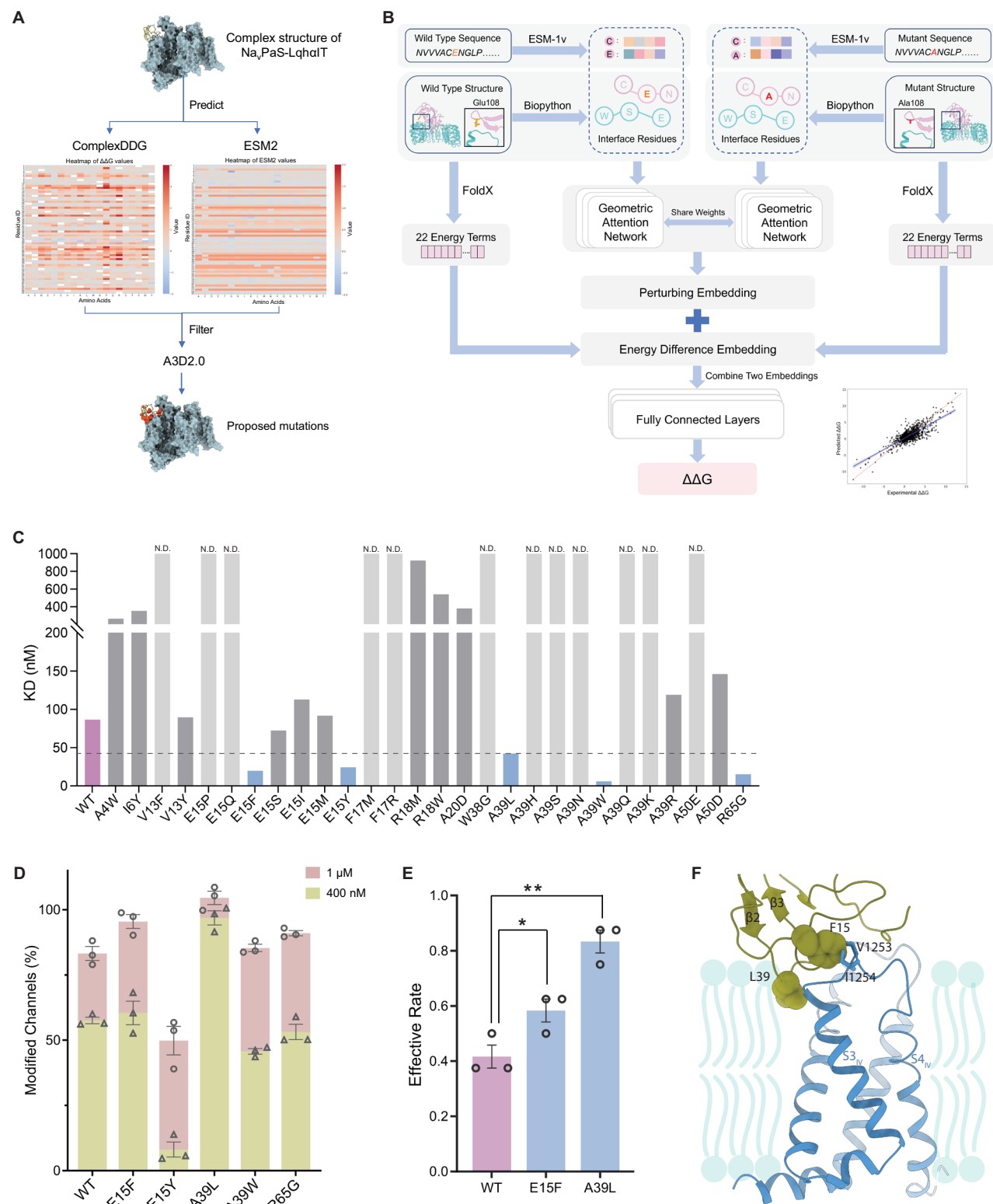

selection based on predicted stability and aggregation propensity. This integrated approach provides a powerful framework for rational toxin engineering and supports the development of next-generation biopesticides with enhanced potency and species selectivity.

Many toxins are known to interact with membrane lipids. Our AI-assisted structure-based design strategy identified LqhαIT A39L as the mutation with the most effective efficacy. Located at the membrane-aqueous interface, Leu39 likely acts as an anchor, stabilizing Na$_v$-LqhαIT interactions. With advancements in AI-based protein design

tools, future developments could go beyond improving the complementarity between peptide toxins and their receptors, expanding to optimize interactions between toxins and the lipid membrane. One promising strategy could involve introducing a hydrophobic loop at a structurally appropriate region of the toxin to enhance such membrane interactions, potentially increasing both binding stability and efficacy.

The synergistic effect of coapplication of toxins that target distinct receptors or different sites on the same receptor is a well-

**Fig. 5 | AI-driven design of LqhαIT-derived biopesticides. A** Flowchart of the mutation-selection process, integrating three core models. Predicted results from ComplexDDG and ESM2 are shown as heatmaps. **B** Overall architecture of ComplexDDG. For each wild-type and mutant complex pair, sequences are embedded with ESM-1v, and 3D structures are parsed with Biopython. Node features for each amino acid in the 3D structural graph are generated by combining its geometric information with ESM-1v sequence embeddings, which are then processed by the weight-shared geometric attention network. The resulting sequence and structure features are concatenated to generate a joint embedding and then fused with the FoldX-calculated energy differences (22 terms) between mutant and wild-type. The resulting comparison feature vector is processed through three fully connected layers to predict the binding ΔΔG, yielding a strong overall correlation between the predicted and experimental values. **C** Affinity of LqhαIT mutants measured by SPR. Data are shown for WT LqhαIT (purple), selected mutations (blue), neutral/weakening mutations (dark gray), and nonbinding mutations (gray). N.D. $K_D$ not detectable. **D** Electrophysiological evaluation of selected LqhαIT mutants on BgNa$_v$1-1a modification at 400 nM and 1 μM. Data are shown as mean ± SEM ($n = 3$ cells). **E** Effective rates of WT LqhαIT and selected mutants in *Galleria mellonella* larvae, measured 24 h post-injection at 1 μg toxin per 100 mg body weight. *$p = 0.024$ for E15F and **$p = 0.0011$ for A39L, compared to WT (one-tailed Student's *t* test). Data are presented as mean ± SEM ($n = 3$ independent replicates, with 8 larvae per replicate). **F** Closed-up views of A39L and E15F at the LqhαIT-Na$_v$PaS interface. Source data are provided as a Source Data file.

established strategy. However, linking toxins that target different binding sites may provide a more efficient approach to developing new channel modulators with enhanced affinity and specificity. This concept is supported by natural toxins with dual domains that target TRPV channels[51]. Our structures offer valuable templates for exploring this approach. Notably, the V4EM and V4EP sites, targeted by LqhαIT and Av3, respectively, are in close proximity, making these two toxins ideal candidates for fusion and optimization. Advanced protein engineering techniques, such as diffusion-based deep learning, could be employed to design and optimize such fusion constructs, potentially yielding highly effective and selective channel modulators. Importantly, however, fusion constructs may introduce unintended effects, such as reduced stability or bioavailability, owing to their increased size. Further investigations are needed to address these potential challenges and optimize the design.

The broader application of peptide-based bioinsecticides has been limited by several key challenges, including poor bioavailability, environmental instability, and delivery constraints. However, the commercial success of SPEAR®, a peptide-based insecticide developed by Vestaron from spider venom, demonstrates that these limitations can be overcome with appropriate formulation strategies. By addressing issues related to stability, bioavailability, and targeted delivery, SPEAR® has achieved practical field efficacy. Building on this progress, future efforts should explore more diverse delivery platforms, such as baculoviral vectors, transgenic expression systems, and nanoparticle-based formulations, to further enhance the stability and effectiveness of peptide-based insecticides in real-world agricultural settings[52-55].

During the preparation of our manuscript, a separate study reported the structure of Na$_v$PaS-LqhαIT[56]. While both studies were conducted independently and reveal broadly similar overall architectures, the resolution of the other published structure is 3.9 Å. This lower resolution results in ambiguous electron densities at the toxin-channel interfaces, hindering accurate modeling of the side chains of key residues, which is critical for elucidating the molecular mechanism of toxin modulation. In contrast, our study presents a higher-resolution structure that provides clearer interaction interfaces, enabling the structure-based design of mutant toxins with enhanced efficacy.

In summary, our study provides valuable insights into the mechanisms of two insect Na$_v$-selective peptide toxins and offers structural templates for bioinsecticide design. By integrating structure-based design with machine learning, we effectively identified beneficial mutations to increase the activity of these peptide toxins. This strategy presents a promising pathway for developing green biopesticides derived from animal toxins that are characterized by high species specificity and efficacy, contributing to sustainable pest management.

## Methods

### Toxin expression and purification
The optimized coding sequences for LqhαIT and Av3 were cloned and inserted into the pLIC-MBP expression vector with the MalE signal sequence, a 6xHis tag, and an MBP fusion tag followed by a tobacco etch virus (TEV) protease recognition site directly preceding the toxin sequence. Toxin mutants were generated via mutagenesis using a QuikChange mutagenesis kit (Tiangen) following the manufacturer's instructions, with primer sequences provided in Supplementary Table 2. The plasmid was transformed into the *E. coli* BL21(DE3) strain (New England Biolabs) for toxin production. The cells were grown in 2xYT medium at 37 °C and induced with 400 μM IPTG when the OD600 reached 0.8. The cells were cultured at 18 °C for an additional 16 h before being collected by centrifugation for 15 min at 5000 × *g*. The cells were resuspended in 10 mM HEPES (pH 7.4) and 250 mM NaCl, disrupted via sonication and centrifuged for 30 min at 12,000 × *g*. The supernatant was loaded onto Ni-NTA resin (GenScript), and the unbound protein was removed by washing with 20 mM imidazole. The target protein was eluted with 500 mM imidazole, followed by concentration and buffer exchange to remove imidazole. One milligram of TEV protease was added to every 40 mg of toxin, and the mixture was incubated at 4 °C for 12 h. The cleaved His6-MBP and His6-TEV were precipitated by the addition of 1% trifluoroacetic acid (TFA), and then the sample was centrifuged at 14,000 × *g* for 30 min. The supernatant was purified via RP-HPLC on a C18 column (Waters, 10 × 250 mm, particle size 5 μm) with a gradient of 20-50% Solvent B (0.05% TFA in acetonitrile) in Solvent A (0.05% TFA in water) over 40 min. The eluted toxin was lyophilized and redissolved in the required buffers. The integrity of the toxins was confirmed via MALDI-TOF and functional bioassays.

### Na$_v$PaS expression and purification
The optimized coding DNA for Na$_v$PaS was cloned and inserted into the BacMam vector with a Twin-Strep tag and a FLAG tag in tandem at the N-terminus. The BacMam viruses were amplified in Sf9 cells according to the standard protocol. HEK293 cells were cultured in Dulbecco's modified Eagle's medium (DMEM) supplemented with fetal bovine serum under 5% CO$_2$ at 37 °C and infected for 48 h before collection. The collected cells were resuspended in 50 mM Tris (pH 7.4) and 150 mM NaCl supplemented with 1% digitonin, 25 μg/ml DNase, protease inhibitor cocktail and 1 mM PMSF and incubated at 4 °C for 2 h. After ultracentrifugation at 150,000 × *g* for 40 min, the supernatant was incubated with anti-FLAG resin (BIMAKE) for 1 h. The resin was washed with wash buffer (50 mM Tris pH 7.4, 150 mM NaCl, 0.06% GDN). The target protein was eluted with wash buffer supplemented with 300 μg/ml FLAG peptide and then applied to Strep-Tactin XT resin. The resin was washed with wash buffer. The protein was eluted with wash buffer supplemented with 50 mM biotin. The eluent was then concentrated and injected into a Superose6 10/300 column (GE Healthcare). For Cryo-EM data collection, the peak fractions were concentrated to ~3 mg/ml and incubated with LqhαIT or Av3 at a 1:4 molar ratio for 30 minutes prior to cryogrid preparation.

### Surface plasmon resonance
SPR experiments were carried out via a Biacore 8 K instrument (GE Healthcare). Na$_v$PaS was immobilized via standard N-hydroxy

succinimide (NHS)/1-ethyl-3-(3-dimethylaminopropyl) carbodiimide hydrochloride (EDC) amine coupling on a carboxyl methyl dextran (CM5) sensor chip (GE Healthcare). Before the covalent immobilization of $Na_v PaS$, the sensor surface was activated by a mixed solution of 0.4 M EDC and 0.1 M NHS (1:1) for 7 min at a flow rate of 10 μl/min. The purified $Na_v PaS$ protein was diluted to 20 μg/ml in immobilization buffer (10 mM sodium acetate, pH 4.5, and 0.04% GDN) and immobilized on the sensor chip to a level of 15,000 response units (RUs). Interactions between $Na_v PaS$ and substrate toxins were monitored by injecting various concentrations of peptides (twofold serial dilutions starting from 400 nM) in running buffer containing 10 mM $Na_2 HPO_4$, 1.8 mM $KH_2 PO_4$, pH 7.4, 137 mM NaCl, and 2.7 mM KCl at a flow rate of 30 μl/min for 120 s. Dissociation was performed by running the buffer without toxins at a rate of 30 μl/min for at least 120 s. The RU was obtained by subtracting a control for unspecific binding (the signal from a blank flow cell). As the inclusion of detergent impaired toxin-channel binding (Supplementary Fig. 5A), all screenings were conducted in detergent-free buffer. The reliability of the assay was confirmed by injecting wild-type LqhαIT as control at the beginning and end of each run, which produced highly consistent responses (Supplementary Fig. 5B, C).

### Expression of BgNa$_v$1-1a and hNa$_v$1.5 in *Xenopus* oocytes
The methodologies for oocyte preparation, cRNA synthesis, and injection were performed as previously described, with slight modifications[57]. Briefly, plasmids (wild-type or mutant) were linearized and purified using phenol-chloroform extraction. cRNA was synthesized in vitro using T7 polymerase and the mMESSAGE mMACHINE High Yield Capped RNA Kit to ensure high-quality and efficient transcription. For optimal functional expression, BgNa$_v$1-1a cRNA was coinjected with TipE cRNA into *Xenopus* oocytes at a 1:1 molar ratio, following an established protocol shown to increase the expression of insect sodium channels in heterologous systems[58]. Similarly, hNav1.5 cRNA was coinjected with the auxiliary β1 subunit into *Xenopus* oocytes at a 1:1 molar ratio to support proper channel expression and functionality.

### Electrophysiological recording and analysis
Sodium currents were measured and analyzed using the two-electrode voltage clamp technique[59]. Recordings were conducted in ND96 bath buffer (2.0 mM KCl, 1.8 mM $CaCl_2$, 96.0 mM NaCl, 10.0 mM 4-(2-hydroxyethyl) piperazine-1-ethanesulfonic acid, and 1.0 mM $MgCl_2$, adjusted to pH 7.5 with NaOH) at 18–24 °C. Glass electrodes were fabricated using a P-1000 puller, filled with a mixture of 3 M KCl and 0.5% agarose, and had resistance of less than 1 MΩ when constructed from borosilicate glass. Sodium currents were recorded using an OC725D oocyte clamp instrument and a Digidata1550A interface. Data acquisition and analysis were performed with pClamp 10.6 software, with the data filtered at 2 kHz and digitized at a 20 kHz sampling frequency. To ensure optimal recording conditions, the amount of cRNA injected into *Xenopus* oocytes and the incubation period were meticulously adjusted, limiting the maximum peak sodium current to ≤2.0 μA. Leak currents were corrected via P/4 subtraction and kept below 0.3 μA. The percentage of BgNa$_v$1-1a modification by LqhαIT and Av3 was calculated using the following Eq. (1):

$$M = \frac{I_{20}}{I_{peak}} \qquad (1)$$

where $I_{peak}$ represents the peak current elicited by a 50 ms step depolarization to −10 mV, $I_{20}$ represents the non-inactivated current at the end of depolarization at 20 ms, and the $I_{20}/I_{peak}$ ratio reflects the efficacy of the toxin in modulating fast inactivation[60]. Similarly, the percentages of hNa$_v$1.5 modified by LqhαIT and Av3 were measured

using the following Eq. (2):

$$M = \frac{I_6}{I_{peak}} \qquad (2)$$

where $I_6$ was used to represents the non-inactivated current at the end of depolarization at 6 ms[33]. $EC_{50}$ values were obtained by fitting the Hill equation to each replicate. Data are presented as the mean ± SEM.

### Insect bioassay
To evaluate the insecticidal effects of the toxins, 5 μL of toxin dissolved in PBS buffer was injected into the second abdominal proleg of last-instar *Galleria mellonella* larvae. Each wild-type and mutant toxin was tested in three independent experiments, with eight larvae per group. As a control, 5 μL of PBS solution was injected into larvae following the same procedure. After injection, the larvae were maintained at room temperature. Death or uncoordinated body movement was recorded as an effective response. Effective rates were calculated 24 h postinjection.

### Cryo-EM sample preparation and data acquisition
For cryo-EM sample preparation, 4 μL of $Na_v PaS$-LqhαIT or $Na_v PaS$-Av3 was loaded on glow-discharged grids (R1.2/1.3, Au 300, Quantifoil). The grids were blotted for 4 s and flash-frozen in liquid ethane cooled with liquid nitrogen using a Vitrobot Mark IV (Thermo Fisher Scientific Inc.) operated at 4 °C and 100% humidity. The grids were subsequently transferred to a Titan Krios electron microscope (Thermo Fisher Scientific, Inc.) operated at 300 kV and equipped with a Gatan K3 Summit direct electron detector and a GIF Quantum energy filter. Movie stacks were automatically collected using an EPU with a preset defocus ranging from −1.2 μm to −1.8 μm in superresolution mode. Data collection was performed at a normal magnification of 81,000× for $Na_v PaS$-LqhαIT and 105,000× for $Na_v PaS$-Av3, with pixel sizes of 1.072 and 0.855 Å/pixel, respectively. The slit width on the energy filter was 20 eV, and the total dose was ~50 e$^-$/Å$^2$ for each micrograph stack.

### Image processing
Image processing was carried out in cryoSPARC[61], and the strategies are shown in Supplementary Fig. 6A and Supplementary Fig. 7A. The movie stacks were motion-corrected via MotionCor2[62], and the defocus values were estimated with patch CTF estimation. Micrographs with contamination or a maximum resolution lower than 5 Å were excluded from the calculation, resulting in a total of 15,651 micrographs ($Na_v PaS$-LqhαIT) and 4515 micrographs ($Na_v PaS$-Av3) for structure determination, respectively.

For the Na$_v$PaS-LqhαIT dataset, a total of 40,338 particles were autopicked via Blob Picker, extracted with a box size of 300 pixels from 500 micrographs, and classified into 50 classes via 2D classification. Representative 2D class averages were selected as the training dataset for Topaz Train[63]. Then, 2,070,023 particles were autopicked from 15,651 micrographs using Topaz. After one round of 2D classification, 1,976,679 particles were selected and then subjected to Ab initio reconstruction in four classes, which were used as templates for the heterogeneous refinement of all the selected particles. After the particles were selected from the most abundant subset, 46.0% of the total particles were subjected to homogeneous refinement, which yielded a reconstruction with a resolution of 3.19 Å. To better resolve the density, another round of heterogeneous refinement was performed, and five volume maps were generated. One density map with 483,207 particles was selected and subjected to nonuniform refinement followed by CTF refinement and nonuniform refinement, improving the map resolution up to 2.87 Å[64,65].

For the Na$_v$PaS-Av3 dataset, a total of 95,454 particles were autopicked via Blob Picker, extracted with a box size of 400 pixels from 200 micrographs, and classified into 50 classes via 2D classification. Representative 2D class averages were selected as the training dataset for

Topaz Train. Then, 1,366,255 particles were autopicked from 4515 micrographs with Topaz. After two rounds of 2D classification, 1,064,365 particles were selected and subjected to ab initio reconstruction in four classes, which were used as templates for the heterogeneous refinement of all the selected particles. After that, the particles selected from the most abundant subset, accounting for 44.0% of the total particles, were subjected to homogeneous refinement, which yielded a reconstruction with a resolution of 3.14 Å. To better resolve the density, another round of heterogeneous refinement was performed, and four volume maps were generated. One density map with 353,627 particles was selected and subjected to nonuniform refinement to improve the resolution of the density to 2.79 Å[64,65].

## Model building and refinement

To generate initial models, the cryo-EM structure of NaᵥPaS (PDB 5X0M) was docked into the Cryo-EM density maps using Chimera[66]. The models were iteratively rebuilt in COOT and refined in Phenix[67,68]. The density maps provided sufficient detail for the reliable modeling of side chains and structural motifs (Supplementary Fig. 8A, B). The statistics for the Cryo-EM data collection and model refinement are reported in Supplementary Table 3. The refined coordinates and Cryo-EM data were deposited into the PDB and EM Data Banks, respectively.

## AI-driven mutation selection

A mutation-selection procedure was conducted using the complex structure of NaᵥPaS-LqhαIT as input. The binding free energy (ΔG) was calculated to evaluate the thermodynamic properties of the interface. The change in protein-protein binding affinity (ΔΔG), representing the difference in binding free energy between the wild-type and mutant complexes, was predicted using ComplexDDG to identify beneficial mutations. Comprehensive details of the ComplexDDG model architecture, benchmarking performance, and ablation studies are provided in the Supplementary Note, Supplementary Figs. 9, 10, and Supplementary Tables 4, 5. Next, the Evolutionary Scale Modeling 2 (ESM2) Large Language Model (pretrained model: esm2_t36_3-B_UR50D) was employed to predict the impact of genetic variations on protein structure and function, facilitating the selection of beneficial mutations[48]. Residue-type probability distributions were computed for each position by masking the residue using ESM2. Additionally, Aggrescan3D (A3D) 2.0 was used to evaluate the potential effects of mutations on protein aggregation[49].

Mutations with predicted ΔΔG scores less than 0 and ESM2 scores greater than those of the wild-type residues were considered candidate beneficial mutations. These were further screened using the A3D 2.0 RESTful service, with the aggregation analysis distance set to 10 Å. A threshold value of <1.5 was applied to minimize the risk of aggregation. To further enrich for functionally favorable variants, a multi-criteria filtering strategy was applied based on structural and physicochemical considerations. Specifically, mutations were prioritized if they: (i) occurred at toxin-channel interface residues identified from the cryo-EM structure; (ii) were predicted to enhance hydrophobic packing or introduce favorable electrostatic interactions; and (iii) did not involve structurally critical residues, such as those involved in disulfide bonds or conserved functional motifs. The resulting candidate mutations were selected for experimental validation.

## Ethical statement

All procedures involving *Xenopus* oocytes were approved by Hainan University Institutional Animal Use and Care Committee and conducted in accordance with institutional guidelines and applicable national regulations of China.

## Reporting summary

Further information on research design is available in the Nature Portfolio Reporting Summary linked to this article.

## Data availability

The cryo-EM maps have been deposited in the Electron Microscopy Data Bank (EMDB) under accession codes EMD-63108 (NaᵥPaS-Av3); and EMD-63189 (NaᵥPaS-LqhαIT). The atomic coordinates have been deposited in the Protein Data Bank (PDB) under accession codes 9LHZ (NaᵥPaS-Av3); and 9LKZ (NaᵥPaS-LqhαIT). The structures used in this paper are available in the PDB database under accession codes 1ANS, 2ASC, 5X0M, 6NT4, 7DTD, 6J8E, 7W77, 6AGF, 7DTC, 8FHD, 7WE4, 7TJ9, and 5XSY. Source data are provided with this paper.

## Code availability

The ComplexDDG code has been deposited in the Zenodo database (https://doi.org/10.5281/zenodo.18150075)[69]. All other computational tools, including ESM-2 and A3D 2.0, are publicly available as described in references[48,49].

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

## Acknowledgements

This research was funded by Fundamental and Interdisciplinary Disciplines Breakthrough Plan of the Ministry of Education of China (no. JYB2025XDXM503 to Z.Y.), the National Natural Science Foundation of China (no. 32372580 to Z.Y., no. 32260666 to S.W. and no. 92156025 to J.M.), the National Key Research and Development Program of China (no. 2025YFC3409400 to Z.Y.), and the Emerging Frontiers Cultivation Program of Tianjin University Interdisciplinary Center (to Z.Y.). We thank Jie Shen from the Tianjin Institute of Industrial Biotechnology, Chinese Academy of Sciences, for assisting with the SPR analysis.

## Author contributions

Conceptualization: H.J., and Z.Y.; Methodology: H.J., R.G., C.W., K.D., F.V.P., and Z.Y.; Investigations: H.J., R.G., C.W., H.X., S.M., Y.G., L.L., X.L., Y.L., L.Y., and R.L.; Resources: Z.Y.; Data analysis: H.J., R.G., C.W., and H.X.; Writing - original draft: H.J., R.G., C.W., H.X., Y.L., and Z.Y.; Writing - review and editing: H.J., J.X., K.D., F.V.P., Z.L., S.W., and Z.Y.; Supervision: Z.L., S.W., and Z.Y.; Project administration: Z.Y.; Funding acquisition: J.M., S.W., and Z.Y.

## Competing interests

The authors declare no competing interests.

## Additional information

¹State Key Laboratory of Synthetic Biology; Frontiers Science Center for Synthetic Biology; Tianjin Key Laboratory for Modern Drug Delivery & High-Efficiency; School of Pharmaceutical Science and Technology, Faculty of Medicine, Tianjin University, Tianjin, China. ²School of Breeding and Multiplication (Sanya Institute of Breeding and Multiplication), Hainan University, Sanya, Hainan, China. ³School of Life and Health Sciences, Hainan University, Haikou, Hainan, China. ⁴School of Tropical Agriculture and Forestry (School of Agricultural and Rural Affairs, School of Rural Revitalization), Hainan University, Danzhou, Hainan, China. ⁵Cryo-electron Microscopy Center and Department of Pharmacology, School of Medicine, Southern University of Science and Technology, Shenzhen Guangdong, China. ⁶MoleculeMind Inc., Beijing, China. ⁷Department of Chemistry, State Key Laboratory Synthetic Biology, Tianjin University, Tianjin, China. ⁸Department of Biology, Duke University, Durham, NC, USA. ⁹Department of Biochemistry and Molecular Biology, Life Sciences Institute, University of British Columbia, Vancouver, BC, Canada. ¹⁰Haihe Laboratory of Sustainable Chemical Transformations, Tianjin, China. ¹¹Guangdong Laboratory for Lingnan Modern Agriculture (Shenzhen Branch), Agricultural Genomics Institute at Shenzhen, Chinese Academy of Agricultural Sciences, Shenzhen Guangdong, China. ¹²These authors contributed equally: Heng Jiang, Ruibo Gao, Huiqin Xu, Cheng Wang. ✉e-mail: liuz3@sustech.edu.cn; wsywsy6000@hainanu.edu.cn; yuchi@tju.edu.cn

