## [Transparent Peer Review file · Nature Communications]

Structural Insights into Insect-Selective Sodium Channel Toxins Drive AI-Enhanced Biopesticide Design

Corresponding Author: Professor Zhiguang Yuchi

Version 0:

Reviewer comments:

Reviewer #1

(Remarks to the Author)

The manuscript "Structural Insights into Insect-Selective Sodium Channel Toxins Drive AI-Enhanced Biopesticide Design" by Heng Jiang is devoted to studies one of the most critical ion channels: voltage gated sodium channel from an insect (*Periplaneta Americana*) sodium channel NavPaS). The significance of this study is obvious to me, since for the first time the details of CryoEM 3D structure of that popular model channel complexed with two aniaml toxins is determined (Av3 from sea anemone and LqhαIT from the scorpion). Great novelty is somehow compromised by the fact, the structure of one of two systems reported has been recently published by an independent research group (Phulera, S., et al., Scorpion alpha-toxin LqhαIT specifically interacts 809 with a glycan at the pore domain of voltage-gated sodium channels. *Structure*, 2024. 32(10): p. 1611-1620 e4). This fact is clearly pointed out in the manuscript. The newer structure of LqhαIT - NavPaS has better resolution which facilitates detailed molecular analysis of interactions and possibly design of new pesticides.

I expect that relatively large community of biophysicist, biologist and some chemists will be interested in presented data. Topic of effective pest control is a serious global issue.

Till now detailed orientations and docking positions of those rater complex peptide toxins systems were a matter of debate, now the issue is resolved. The molecular details plus AI based analysis of promising mutation positions in scorpion toxin helped to design new more potent and hopefully also selective anti-insect peptides. The synthesis of those new compounds revealed their postulated efficacy. This is a direct proof that AI-assisted improvement of peptide-protein binding strength is possible. Such approach not only suggest new anti-insect effective compounds but offers opportunity to use similar pipelines in rational AI-assisted drug design. This refers to situation when a potent inhibitors of known protein targets are sought. I rank this manuscript high, it contains a lot of new facts regarding two important molecular complexes, with significant impact on economy and possibly human health. The methods used are divers and link technical expertise with goal focused experiments. I am not in position to judge how appropriate were experimental details regarding proteins expression, purification and mutagenesis, but structural studies and analysis seems to be fine.

The quality and clarity of presentation are very good. Conclusions are supported by data. I have only a few critical comments:

1. Using different sodium channels for electrophysiology and other one for all structural studies is somehow risky and not common. Perhaps adding event more comments on the utility of such an approach (?previous studies) would better justify that part of the manuscript (we understand argument on lack of conductance of NavPaS in used cell system, but may be other arguments might facilitate transferring measurements results on orthologous Nav channel BgNav1-1a, from the German cockroach *Blattella germanica* to NavPaS.
2. Line 148: "BgNav1-1a shares approximately 59% sequence identity with NavPaS and is well suited for electrophysiological studies [36, 37]. As a control, we used the cardiac isoform of the human Nav channel (hNav1.5), which represents mammalian Nav channels."
How much sequence identity is shared between NavPaS and hNav1.5?
3. Line 71 : calmodulin?
4. Line 165-166, not quite precise. The authors may add some review comments on results of molecular docking/MD of both toxins to NavPaS made in the past, for example: (1) Scorpion Toxins: Positive Selection at a Distal Site Modulates

- Functional Evolution at a Bioactive Site, Limei Zhu, Bin Gao, Shouli Yuan, Shunyi Zhu; *Mol Biol Evol.* 2019 Feb 1;36(2):365-375. doi: 10.1093/molbev/msy223; (2) Niklas, B. et al. Interactions of Sea Anemone Toxins with Insect Sodium Channel—Insights from Electrophysiology and Molecular Docking Studies. *Molecules* 26, 1302 (2021), etc.
5. Lines 239-241. “Thus, mutations at these residues could disrupt interactions with both the Nav channel and lipid membrane, emphasizing the critical role of membrane interactions in Av3 activity.” – in my opinion this statement is quite speculative and too strong. Is interaction with a membrane CRITICAL in Av3 activity?
 6. Line 348 “we applied manual filtering based on empirical knowledge and included substitutions targeting key sites identified by ComplexDDG” – I am concerned about substantial subjective unspecified judgement revealed in this point. Could authors’ results be reproduced?
 7. Line 423. It should be indicated here in Discussion section in what molecule this A39L mutation is.
 8. Numerous references deserve better editing, pages range is missing, for example [2], [13], [18], [29-33]. Authors are not properly shown in [15].
 9. A general remark: What is the translation of the toxin's effect on fast inactivation from the NavPas channel to other insect Nav channels, given that NavPas is non-functional, has no inactivation particle IFM (MFM) motif, and it is unknown whether it preserves the fast inactivation process?
 10. May be authors should be more critical about prospects of animal derived toxins in pest control. Those toxins have been known for decades and have failed to market as their excellent in vitro effects do not translate into in vivo effects.
 11. Is computational pipeline to design new variants of toxins available for interested scientists?

Reviewer #2

(Remarks to the Author)

This is a well-executed and timely study that provides valuable structural and functional insights into insect-selective sodium channel toxins. The authors effectively integrate cryo-EM, surface plasmon resonance, and electrophysiology to elucidate how Av3 and Lqh α IT differentially engage NavPaS, modulate channel gating, and confer insect-specific potency at the structural level. The discovery of a previously uncharacterized membrane-embedded binding site for Av3 is particularly compelling, as it may open new opportunities for rational design targeting this unique pocket. Furthermore, the application of AI-guided protein engineering led to a redesigned Lqh α IT variant with significantly enhanced potency. These findings not only deepen our understanding of toxin–channel coevolution but also establish a strong framework for the development of next-generation biopesticides with improved specificity and efficacy. There are still a few issues that should be addressed, but if properly resolved, I would support publication.

Major points:

Line 90: The authors state that ‘in nature, many animal toxins have evolved to selectively target insect Nav channels.’ Do the authors have a hypothesis for why this evolutionary pressure favored insect-selective targeting? Why wouldn’t similar toxins evolve to target mammalian Nav channels instead?”

Line 147: The authors used NavPaS for structural studies, whereas BgNav1-1a was employed for electrophysiological analyses. Why not use BgNav1-1a for the cryo-EM studies as well? Doing so could provide a more direct correlation between structure and function.”

Line 150: Is there a specific reason the authors chose human Nav1.5 as a surrogate for other human Nav channels? It would be helpful if the authors could elaborate further—based on the resolved structures—on whether the observed selectivity toward Nav1.5 might also extend to other human Nav isoforms.

Line 222: The authors state that ‘key residues at the VSD4 interface, including Pro5, Tyr7, Trp13, and Tyr18, are crucial for Av3 binding.’ However, based on Fig. 3c and the provided PDB file, Pro5 does not appear to participate in any direct interaction with the channel. Furthermore, the authors state, ‘Alanine substitutions at these residues nearly abolish binding, as confirmed by surface plasmon resonance (SPR) (Supplementary Table 2)’. This appears inconsistent with the data (Supplementary Table 2), as the P5A mutation shows a binding affinity similar to or slightly better than the wild-type (KD: 20.4 nM vs. 21.5 nM). The authors should revise this statement to accurately reflect the experimental results.”

Line 274-277, 284-286: Could the authors provide a more detailed discussion of the mechanism underlying the significant conformational shifts of Glu15, Phe17, and Arg18?

Figure 3H: Since the authors state in the legend that invertebrate-specific residues are marked with an asterisk, it is confusing that M281 is marked this way, suggesting it is invertebrate-specific. However, based on Figure 3H, M281 appears to be conserved across both invertebrate and mammalian Nav channels, except for hNav1.8, where it is a lysine (K). This should be clarified.

Additionally, in the main text (lines 253–257), the authors state: ‘The selectivity is attributed primarily to invertebrate-specific residues in PD1, including Ile279, Val283, Trp391, His392, and Leu394 (Fig. 3H). Substituting these invertebrate-specific residues with their mammalian counterparts significantly reduces the efficacy of Av3 by 2.6- to 44-fold, underscoring their importance in species selectivity (Fig. 3G).’ However, Figure 3G does not include data for the Ile279 and Leu394 mutation. The authors should either provide the corresponding data or revise the text to accurately reflect the figure contents.

Minor points:

Line 204-206: Add reference for this statement 'This mechanism aligns with the structural changes induced by the toxin AaH2, further supporting a common role of electrostatic interactions in toxin-induced channel modulation.'

Line 226, line 256, line 887, line 899: The authors should clarify which assay was used to measure Av3 activity here, in order to distinguish it from the previously mentioned SPR assay."

Line 239: The authors state that 'mutations at these residues could disrupt interactions with both the Nav channel and lipid membrane,' which implies that this is a speculative statement. However, Figure 3F clearly demonstrates that the W8A and W13A mutations substantially reduced activity. The authors should revise this sentence to reflect the experimental evidence rather than presenting it as a hypothesis.

Line 244: The reference and PDB ID for the apo structure of Av3 should be provided.

Line 271: The authors should add the data source (figure or table) supporting the statement '... LqhαIT activity was reduced by 40-fold and 37-fold, respectively.'

Line 304: The author should also discuss the amino acid differences in S3-S4 linker between insect and mammal sodium channels.

Line 309: The authors should provide reference and/or figure source to support the statement 'While the glycans near the glycosylation site are relatively conserved across Nav channels, their distal regions vary'

Line 585, Line 620: $50 \text{ e}^-/\text{Å}^2$ should be $50 \text{ e}^-/\text{Å}^2$. 3,53,627 particles should be 353,627 particles.

Line 711: The journal name Annual Review of Entomology is not in the abbreviated format. Please review all references to ensure consistent formatting throughout the reference list.

Line 920-924: The authors showed EeNav1.4 in supplementary Fig.4 but did not mention it in legend.

Line 950-951: The authors should clarify the meaning of 'N.D.' to avoid ambiguity.

Figure 1A: To provide a more complete picture of the electrophysiological effects, the authors should show representative current traces of Av3 and LqhαIT on hNav1.5.

Figure 1B: The hNav1.5 curve for Av3 appears relatively smooth, whereas the curve for LqhαIT shows large error bars. The authors should provide an explanation for this discrepancy. In addition, the figure legend should clearly indicate whether the data are presented as mean ± SD or mean ± SEM, and the number of replicates (n) should be specified."

Figure 2B: The authors should indicate what the dashed and solid lines represent in the right panel for clarity.

Figure 3E: The authors should clarify how the lipid was identified as a phospholipid based on the density map. Additionally, the sigma value used for contouring the map should be specified in the figure legend.

Figure 3F, Figure 3G, Figure 4F, Figure 4G, Figure 5B, Figure 5C, Figure 5D: To enhance transparency and allow better interpretation of data variability, the authors are encouraged to display individual replicate values and specify the number of replicates (n) in the bar plots.

Figure 3H: D1 or DI? Please be consistent.

Figure S3A and S3D: The text '86 residues' appears to be unintended and should be removed.

Table-S2: The authors also performed SPR analysis for LqhαIT variants, but this was not mentioned in the main text. Is there a specific reason for this omission?

Reviewer #3

(Remarks to the Author)

The structural basis for the action of peptide toxins acting selectively on insect sodium channels is poorly understood but can provide valuable information for the future design of effective and selective insecticides. In this study, Jiang and colleagues determined cryo-EM structures of the cockroach sodium channel NavPs in complex with either the LqhaIT or Av3 insecticidal toxin. The binding interfaces between toxin and channel were probed using mutagenesis, which included introducing mammalian substitutions into the insect channel. Mutants were tested using surface plasmon resonance studies with NavPs or two-electrode voltage-clamp studies with the BgNav1.1 channel from a different cockroach species (as NavPs is non-functional). This study also describes an AI modelling approach used to identify mutations of LqhaIT that were predicted to increase its binding affinity. One such mutant (A39L) was shown to be twice as effective as the wildtype toxin in a bioassay.

This is very interesting work that would be of great interest to ion channel pharmacologists, particularly those seeking to develop novel insecticides. One exciting finding is how different the Av3 binding site is from that of LqhaIT (as the sites have very little overlap) and how Av3 is buried at an interface between the VSD4 and pore domain. Overall, the cryo-EM work is well described and has clear, attractive figures and the electrophysiology work also clear and straight-forward. There are a number of issues, though, that the authors should address, particularly regarding the surface plasmon resonance (SPR) work.

Lines 224-225 describe how alanine mutations of Av3 'nearly abolish binding' but that some of these alanine mutations produce only a ~2 fold reduction in Av3 activity when tested using electrophysiology. Leaving aside the issue that two different cockroach channels were used (NavPs vs BgNav1.1), it seems that the NavPs immobilisation and subsequent toxin binding steps of SPR were performed with buffers lacking detergent. This membrane protein is almost certainly to have undergone some unfolding in the absence of detergent and, although some part of the receptor site may remain to allow toxin binding, the authors cannot have confidence that NavPs is adopting a native conformation and that the results are therefore physiologically relevant. This is also an issue for the AI approach to mutant generation, as SPR was the experimental technique used to first screen AI-suggested mutations of LqhaIT (Fig 5B). I suspect the use of SPR limited the effectiveness of this approach and that valid affinity-increasing mutants were overlooked. Can the authors comment on this and, if detergent was in fact included in the buffers, please provide example sensorgrams?

Lines 195-206 describe how toxin-induced conformational changes generate binding contacts between different sections of the channel: the DIII-DIV linker, DIV S6 helix and C-terminal domain. The issue is that the amino acids involved in salt bridges & hydrogen bonds differ between NavPs and the BgNav1.1 & Nav1.5 channels: NavPs R1293 is alanine in BgNav1.1 and methionine in Nav1.5; NavPs R1138 is lysine in BgNav1.1 and asparagine in Nav1.5; NavPs D1420 is glutamate in Nav1.5. It would be helpful to show a sequence alignment of these channel sections in Fig 2 and for the authors to point out that interactions shown for the NavPs structure will differ in other channels.

Lines 257-259 speculate that "variations in glycosylation patterns may influence toxin selectivity" and, in lines 309-312, that the distal regions of glycans vary. While this is an intriguing suggestion, the authors should provide a reference(s) that describes differences in glycosylation between insects and mammals. There is also the issue that LqhaIT mutants with substitutions from the mammalian-selective AaH2 toxin were tested electrophysiologically in amphibian (*Xenopus*) cells and so the claim based on results shown in Fig 4F that interactions between LqhaIT and the glycan "highlights the critical role of this region in determining selectivity" is highly speculative.

'N=' values should be reported throughout e.g. in figures with electrophysiology and bioassay data.

MINOR POINTS:

Lines 41-43: It is not clear why the Av3-bound channel is proposed to be in a 'deactivated state' while the LqhaIT-bound channel has 'disrupted fast inactivation'. The ion-conducting pores of apo-, Av3- and LqhaIT structures look identical (sup Fig 3C, F) and both toxins inhibited fast inactivation. It doesn't make sense to distinguish the two toxins as one caused deactivation and the other causing disrupted fast inactivation.

L47: "remarkable doubling in efficacy". It might help to point out that this was tested by bioassay

L71: "calmoduzlin" ... calmodulin

L201: "Arg1277-Glu1435". It might be better to change Arg1277 to 'R6' to be consistent with the figure. Alternatively, "Arg1277 (R6)-Glu1435" could work

L202: "binding results in only the first two bridges" ... this is not specific enough, especially as three bridges are shown in Fig 2B

L202: "Pro5, Tyr7, Trp13, and Tyr18, are crucial for Av3 binding (Fig. 3C). Alanine substitutions at these residues nearly abolish binding, as confirmed by surface plasmon resonance (SPR)". The SPR results for Y7, W13 and Y18 are "N.D.", which stand for 'not determined', indicating no data was collected on them. In addition, the SPR data shows the P5 mutant has a similar affinity as wildtype. These two sentences therefore need clarification.

L230: "reduce the binding affinity and efficacy of Av3, with Trp8 ... having more pronounced effect". SPR data shows 'N.D.' for W8 and so binding affinity wasn't determined.

L280: "the D1252R mutation completely abolishes it (Fig 4C, G)". The D1252R mutation is not shown in Fig 4G.

L311: "Fig. 4E, H, F" ... I believe this should cite Fig 4B instead of Fig 4E.

L348: "we applied manual filtering based on empirical knowledge". This requires further explanation i.e. what empirical knowledge were you acting on?

L414: "The distal region of the glycosylation site at Asn330 has diverged during evolution (Supplementary Fig. 4)". Sup Fig 4 shows the glycan from a number of sodium channel cryo-EM structures. However, all of these channels (or the majority,

including NavPs) were expressed in the mammalian HEK cell system and so Sup Fig 4 cannot be used to support this statement about glycosylation differences during evolution.

L860: "R3 crossing the hydrophobic constriction site". Please check as I believe it is R4 that crosses the HCS.

L869: "superimposed onto with its apo state" ... no need for the word 'with'

L877: "invertebrate-specific residues are marked with an asterisk". Some residues are inappropriately marked with an asterisk as they are conserved between insect and most/all-but-one mammalian channels. Please be more selective in the use of asterisks to mark invertebrate-specific residues.

L892: "invertebrate-specific with an asterisk" ... please be more selective in the use of asterisks to mark invertebrate-specific residues.

L929: "Bar chart ranking prediction models, showing the superior accuracy of ComplexDDG". There is insufficient detail here or in Methods or elsewhere in the text describing how this comparison between different softwares was performed.

Sup Fig 3C, F. The side chains shown in the pore should be identified and the reason for showing them stated.

Version 1:

Reviewer comments:

Reviewer #2

(Remarks to the Author)

The authors have adequately addressed all of my previous concerns. I recommend acceptance of the manuscript in its current form.

Reviewer #3

(Remarks to the Author)

The authors have addressed all my concerns and their changes have strengthened the manuscript.

One of my major concerns was about the SPR studies carried out in the absence of detergent. In their reply to reviewers, the authors have made a clear case for why they performed their SPR studies without detergent and have supplied a convincing figure with sensograms showing (a) that detergent interferes with toxin binding and (b) that binding curves of wildtype Lqh α IT "at the beginning and the end of a screening experiment" look almost identical. My confidence in the SPR data has consequently greatly increased. I believe it would be beneficial for readers (and especially those planning to replicate this approach with their own toxin/binder of interest), that this figure is added as a supplementary figure to the manuscript. A sentence or two should be added to the Results about the negative effect of detergent on ligand binding and that control toxin bindings were carried at the beginning and end of a screening experiment to verify integrity of the receptor site. If the immobilised channel was washed with buffer containing detergent in between toxin binding runs then this would be an important point to also add.

Reviewer #4

(Remarks to the Author)

The discussion of their self-developed "ComplexDDG" algorithm is not convincing and may need to be improved.

1. There is no detailed (or even a general) discussion of what are the main idea, method, architecture, or innovation of the newly proposed "ComplexDDG" model. It is totally unclear the performance of this model is due to model innovation, pretrain knowledge, or even just some leaking of the data. Without a good discussion of the details, it is impossible to properly evaluate the performance of the model.

2. There are various mistakes in the comparison with state-of-the-art models. For instance, the authors compare their model with "TopNetTree" and "TopGBT" models and they cite the reference 51,

McMurray HR, et al. Gene network modeling via TopNet reveals functional dependencies between diverse tumor-critical mediator genes. Cell Rep 37, 110136 (2021)

It is obvious that this is the wrong reference. The "TopNet" model in ref 51 IS NOT the "TopNetTree" and "TopGBT" models for protein-protein binding affinity change upon mutation. In fact, the right reference should be Menglun Wang, Z. X. Cang, and Guo-Wei Wei, A topology-based network tree for the prediction of protein-protein binding free energy changes following mutation, Nature Machine Intelligence, 2, 116-123 (2020).

3. It seems that the comparison results in "Supplementary Fig. 5 Performance of the ComplexDDG model" are adopted from supplementary files of reference 53

Shan S, et al. Deep learning guided optimization of human antibody against SARS-CoV-2 878 variants with broad neutralization. Proc Natl Acad Sci U S A 119, e2122954119 (2022).

In which, the results of "TopNetTree" and "TopGBT" models are significantly lower than all existing models. In fact, it is totally unclear how the authors in PNAS paper get the results, considering that they also cite the wrong reference in their table (Note that they cite ref 46 in their Table, but ref 46 has nothing to do with "TopNetTree" and "TopGBT" models).

Version 2:

Reviewer comments:

Reviewer #3

(Remarks to the Author)

The authors have made the requested changes to the manuscript and I recommend its publication.

Reviewer #4

(Remarks to the Author)

All my concerns are well addressed! I have no further comments and would recommend the publication of the paper!

Dear Reviewers,

We thank all the reviewers for the constructive comments and suggestions, which have led to an overall improvement in our efforts to communicate the results. I have reproduced the reviewers' comments verbatim below together with our responses. For clarity, criticisms and/or suggestions from the reviewers are colored in dark red, while our response is shown in dark blue.

Reviewer #1 (Remarks to the Author):

The manuscript "Structural Insights into Insect-Selective Sodium Channel Toxins Drive AI-Enhanced Biopesticide Design" by Heng Jiang is devoted to studies one of the most critical ion channels: voltage gated sodium channel from an insect (*Periplaneta Americana*) sodium channel NavPaS). The significance of this study is obvious to me, since for the first time the details of CryoEM 3D structure of that popular model channel complexed with two animal toxins is determined (Av3 from sea anemone and LqhαIT from the scorpion). Great novelty is somehow compromised by the fact, the structure of one of two systems reported has been recently published by an independent research group (Phulera, S., et al., Scorpion alpha-toxin LqhαIT specifically interacts 809 with a glycan at the pore domain of voltage-gated sodium channels. *Structure*, 2024. 32(10): p. 1611-1620 e4). This fact is clearly pointed out in the manuscript. The newer structure of LqhαIT - NavPaS has better resolution which facilitates detailed molecular analysis of interactions and possibly design of new pesticides.

I expect that relatively large community of biophysicist, biologist and some chemists will be interested in presented data. Topic of effective pest control is a serious global issue.

Till now detailed orientations and docking positions of those rater complex peptide toxins systems were a matter of debate, now the issue is resolved. The molecular details plus AI based analysis of promising mutation positions in scorpion toxin helped to design new more potent and hopefully also selective anti-insect peptides. The synthesis of those new compounds revealed their postulated efficacy. This is a direct proof that AI-assisted improvement of peptide-protein binding strength is possible. Such approach not only suggest new anti-insect effective compounds but offers opportunity to use similar pipelines in rational AI-assisted drug design. This refers to situation when a potent inhibitors of known protein targets are sought.

I rank this manuscript high, it contains a lot of new facts regarding two important molecular complexes, with significant impact on economy and possibly human health. The methods used are divers and link technical expertise with goal focused experiments. I am not in position to judge how appropriate were experimental details regarding proteins expression, purification and mutagenesis, but structural studies and analysis seems to be fine.

The quality and clarity of presentation are very good. Conclusions are supported by data. I have only a few critical comments:

COMMENT #1: Using different sodium channels for electrophysiology and other one for all structural studies is somehow risky and not common. Perhaps adding event more comments on the utility of such an approach (? previous studies) would better justify that part of the manuscript (we understand argument on lack of conductance of NavPaS in used cell system,

but may be other arguments might facilitate transferring measurements results on orthologous Nav channel BgNav1-1a, from the German cockroach *Blattella germanica* to NavPaS.

RESPONSE #1: We thank the reviewer for this thoughtful comment. To better justify our use of BgNav1-1a for functional validation of the non-conducting NavPaS cryo-EM structures, we have cited a previous study by Nieng Yan's group, which similarly used NavPaS for structural characterization and BgNav1-1a for electrophysiological analysis in the context of the Dc1a toxin²⁹. Although NavPaS and BgNav1-1a share ~59% overall sequence identity, the most divergent regions are in the intracellular loops and C-terminal domain. Importantly, the extracellular domains that interact with toxins are more conserved between the two channels, which supports the use of BgNav1-1a as a functional proxy. We have added the following text to the Results section (Line 150-155) to clarify this point: "Therefore, for electrophysiological analyses, we employed the orthologous insect Nav channel, BgNav1-1a, from the German cockroach *Blattella germanica*, which was also used in a previous study to functionally validate the NavPaS-Dc1a complex structure²⁹. BgNav1-1a shares approximately 59% sequence identity with NavPaS, with higher conservation in the extracellular toxin-interacting regions, making it a suitable surrogate for electrophysiological studies^{36, 37}."

COMMENT #2: Line 148: "BgNav1-1a shares approximately 59% sequence identity with NavPaS and is well suited for electrophysiological studies [36, 37]. As a control, we used the cardiac isoform of the human Nav channel (hNav1.5), which represents mammalian Nav channels."

How much sequence identity is shared between NavPaS and hNav1.5?

RESPONSE #2: NavPaS shares approximately 42% overall sequence identity with hNav1.5. We have now added this information to the revised manuscript (Line 155-158): "As a control, we used the cardiac isoform of the human Nav channel (hNav1.5), which provides preliminary assurance of mammalian safety. hNav1.5 is representative of mammalian Nav channels and shares 42% overall sequence identity with NavPaS."

COMMENT #3: Line 71: calmodulin?

RESPONSE #3: We have corrected the typographical error from "calmoduzlin" to "calmodulin" in the revised manuscript.

COMMENT #4: Line 165-166, not quite precise. The authors may add some review comments on results of molecular docking/MD of both toxins to NavPaS made in the past, for example: (1) Scorpion Toxins: Positive Selection at a Distal Site Modulates Functional Evolution at a Bioactive Site, Limei Zhu, Bin Gao, Shouli Yuan, Shunyi Zhu; *Mol Biol Evol.* 2019 Feb 1;36(2):365-375. doi: 10.1093/molbev/msy223; (2) Niklas, B. et al. Interactions of Sea Anemone Toxins with Insect Sodium Channel—Insights from Electrophysiology and Molecular Docking Studies. *Molecules* 26, 1302 (2021), etc.

RESPONSE #4: We thank the reviewer for this helpful suggestion. Several previous studies, including molecular docking and mutagenesis analyses, have suggested that Av3 and LqhαIT target regions near the classical site 3 on insect sodium channels. Our structural data refine and extend these observations: while LqhαIT indeed binds at the classical extracellular site on VSD4, Av3 inserts more deeply into a cleft between VSD4 and PD1, occupying a previously

uncharacterized membrane-embedded binding site. We have incorporated the suggested references and revised the relevant paragraph in the manuscript as follows: (Line 167-174) “Previous studies have shown that Av3 competes with other site 3 toxins, suggesting it targets the same or overlapping regions within VSD4³⁸. This was further supported by docking studies showing that Av3 sits on top of VSD4³⁹. Mutagenesis experiments have also indicated that certain site 3 mutations differentially affect Av3 and scorpion toxins, implying distinct binding modes, although the precise binding pose of Av3 remains unclear²⁵. Our cryo-EM structure of NavPaS-Av3 reveals that Av3 occupies a previously uncharacterized membrane-embedded site on Nav channels.”, and (Line 180-183) “In contrast, LqhαIT binds to the extracellular side of VSD4, interacting with the glycan moiety from PD1 and positioning itself above the membrane (Fig. 1D, 4A). This binding site closely resembles the region targeted by other known α-ScTxs, including AaH2, Lqh3, and MTα-5^{30, 33, 34} (Supplementary Fig. 1).”

COMMENT #5: Lines 239-241. “Thus, mutations at these residues could disrupt interactions with both the Nav channel and lipid membrane, emphasizing the critical role of membrane interactions in Av3 activity.” – in my opinion this statement is quite speculative and too strong. Is interaction with a membrane CRITICAL in Av3 activity?

RESPONSE #5: We thank the reviewer for this valuable comment. To clarify and better support our interpretation, we have incorporated two additional lines of evidence: 1) In our cryo-EM structure, Av3 interacts with NavPaS by sandwiching an explicit lipid molecule, which is positioned between the toxin and the channel. This lipid is consistently observed in several human Nav structures, including hNav1.3 (PDB: 7W77), hNav1.7 (PDB: 8F0Q), and hNav1.8 (PDB: 9DBK), at an equivalent position, suggesting it is a structurally stable component of the channel environment. 2) Alanine substitutions at two lipid-facing tryptophan residues in Av3 (Trp8 and Trp13) significantly reduced the toxin's binding and activity, indicating that these interactions contribute to functional engagement with the channel. We have accordingly revised the manuscript to adopt a more cautious tone. The revised text now reads (Line 249-259): “Additionally, some hydrophobic residues in Av3, such as Trp8 and Trp13, face the toxin-lipid interface in our complex structure, suggesting that they may help stabilize the toxin conformation through interactions with the lipid membrane. A well-defined lipid-like density was observed near Trp8 of Av3 and Trp391 of PD1 (Fig. 3E). Based on its shape and position, analogous to lipids observed at equivalent sites in other human Nav structures, it was modeled as a putative phospholipid, suggesting a conserved lipid-channel interaction⁴¹⁻⁴³. Notably, Av3 does not displace the lipid but rather ‘leans’ on it, forming a sandwich-like arrangement with NavPaS. Electrophysiological assays showed that substitution of Trp8 and Trp13 with alanine markedly reduced Av3 activity (Fig. 3F), supporting a contributing role of membrane interactions in Av3 function.”

COMMENT #6: Line 348 “we applied manual filtering based on empirical knowledge and included substitutions targeting key sites identified by ComplexDDG “ – I am concerned about substantial subjective unspecified judgement revealed in this point. Could authors’ results be reproduced?

RESPONSE #6: We thank the reviewer for raising this important concern regarding potential subjectivity in our mutation selection process. To clarify, the “empirical knowledge-based filtering” was guided by reproducible structural and biophysical criteria rather than subjective judgment. Specifically, we applied a multi-criteria strategy to prioritize substitutions that met the

following conditions: 1) Proximity to the toxin-Nav interface, as defined by our cryo-EM structure; 2) Predicted improvement in hydrophobic packing or electrostatic complementarity, based on side chain properties; 3) Preservation of structural integrity, by avoiding mutations at residues involved in disulfide bonds or key secondary structure elements. Additionally, we included several rational substitutions at sites identified by AI-based tools, further expanding the mutation space. These criteria and procedures have now been described in detail in the revised Methods section to ensure reproducibility. The relevant text now reads (Line 719-725): “To further enrich for functionally favorable variants, a multi-criteria filtering strategy was applied based on structural and physicochemical considerations. Specifically, mutations were prioritized if they: (i) occurred at toxin-channel interface residues identified from the cryo-EM structure; (ii) were predicted to enhance hydrophobic packing or introduce favorable electrostatic interactions; and (iii) did not involve structurally critical residues, such as those involved in disulfide bonds or conserved functional motifs.”

COMMENT #7: Line 423. It should be indicated here in Discussion section in what molecule this A39L mutation is.

RESPONSE #7: The A39L mutation refers to a substitution in LqhαIT. To avoid any ambiguity, we have revised the text in the Discussion section to explicitly state “LqhαIT A39L”.

COMMENT #8: Numerous references deserve better editing, pages range is missing, for example [2], [13], [18], [29-33]. Authors are not properly shown in [15].

RESPONSE #8: We have thoroughly reviewed and revised all references to ensure completeness and accuracy, including adding missing page ranges and correcting author information where necessary.

COMMENT #9: A general remark: What is the translation of the toxin's effect on fast inactivation from the NavPaS channel to other insect Nav channels, given that NavPaS is non-functional, has no inactivation particle IFM (MFM) motif, and it is unknown whether it preserves the fast inactivation process?

RESPONSE #9: We thank the reviewer for this thoughtful comment. We fully acknowledge that NavPaS is a non-conducting channel lacking the canonical fast inactivation motif. However, NavPaS remains the only insect Nav channel to date that is structurally tractable for high-resolution cryo-EM analysis. To address the functional relevance, our key conclusions regarding toxin-induced modulation of fast inactivation are derived from electrophysiological recordings using BgNav, a functional *para*-type insect Nav channel that contains the conserved inactivation machinery and displays robust fast inactivation kinetics. BgNav is representative of Nav channels found in major insect pests and is widely used in functional studies. Our recordings clearly show that both Av3 and LqhαIT impair fast inactivation in BgNav, and that this effect is lost upon mutation of key toxin-channel interface residues, supporting a conserved functional mechanism. Structurally, the NavPaS-toxin complexes reveal that toxin binding traps the S4 segment of VSD4 in a downward conformation. Since fast inactivation is allosterically coupled to upward S4 movement, this stabilization is consistent with the impaired fast inactivation observed functionally in BgNav. Thus, while NavPaS itself lacks fast inactivation, the conserved domain architecture and toxin-induced conformational changes offer valuable mechanistic

insights. In short, although NavPaS is non-functional, it remains a useful structural surrogate for mapping toxin interactions with *para*-type insect Nav channels. Given the evolutionary conservation of channel architecture and functional modulation by site-3 toxins, the mechanism elucidated here is likely broadly applicable and may aid in the structure-based design of insecticides targeting fast inactivation pathways in insect Navs.

COMMENT #10: May be authors should be more critical about prospects of animal derived toxins in pest control. Those toxins have been known for decades and have failed to market as their excellent *in vitro* effects do not translate into *in vivo* effects.

RESPONSE #10: We thank the reviewer for this important and balanced perspective. We agree that many animal-derived toxins, despite demonstrating strong *in vitro* efficacy, have historically struggled to achieve *in vivo* success due to critical limitations such as poor bioavailability, environmental instability, and inefficient delivery. These challenges have indeed hindered their translation into viable pest control agents. However, recent advances suggest that these barriers are increasingly surmountable. A notable example is SPEAR[®], a commercially approved peptide-based insecticide developed by Vestaron from spider toxin, which demonstrates that with appropriate formulation and delivery strategies, animal-derived toxins can achieve practical field efficacy. Furthermore, emerging platforms, including baculoviral vectors, transgenic crop expression, and nanoparticle-based delivery systems, offer promising solutions to enhance the stability, targeting, and bioavailability of peptide-based bioinsecticides. We have added a corresponding paragraph to the Discussion to critically address this issue (Line 504-513): “The broader application of peptide-based bioinsecticides has been limited by several key challenges, including poor bioavailability, environmental instability, and delivery constraints. However, the commercial success of SPEAR[®], a peptide-based insecticide developed by Vestaron from spider venom, demonstrates that these limitations can be overcome with appropriate formulation strategies. By addressing issues related to stability, bioavailability, and targeted delivery, SPEAR[®] has achieved practical field efficacy. Building on this progress, future efforts should explore more diverse delivery platforms, such as baculoviral vectors, transgenic expression systems, and nanoparticle-based formulations, to further enhance the stability and effectiveness of peptide-based insecticides in real-world agricultural settings⁵⁸⁻⁶¹.”

COMMENT #11: Is computational pipeline to design new variants of toxins available for interested scientists?

RESPONSE #11: We thank the reviewer for the interest in our computational pipeline. The ComplexDDG algorithm, developed by our commercial partner MolecularMind, is currently not publicly available. However, efforts are ongoing to make the platform accessible to the broader scientific community in the future. In the meantime, other components of our computational workflow, including A3D2.0 and ESM2, are publicly available and have been appropriately cited.

Reviewer #2 (Remarks to the Author):

This is a well-executed and timely study that provides valuable structural and functional insights into insect-selective sodium channel toxins. The authors effectively integrate cryo-EM, surface

plasmon resonance, and electrophysiology to elucidate how Av3 and LqhαIT differentially engage NavPaS, modulate channel gating, and confer insect-specific potency at the structural level. The discovery of a previously uncharacterized membrane-embedded binding site for Av3 is particularly compelling, as it may open new opportunities for rational design targeting this unique pocket. Furthermore, the application of AI-guided protein engineering led to a redesigned LqhαIT variant with significantly enhanced potency. These findings not only deepen our understanding of toxin–channel coevolution but also establish a strong framework for the development of next-generation biopesticides with improved specificity and efficacy. There are still a few issues that should be addressed, but if properly resolved, I would support publication.

Major points:

COMMENT #1: Line 90: The authors state that ‘in nature, many animal toxins have evolved to selectively target insect Nav channels.’ Do the authors have a hypothesis for why this evolutionary pressure favored insect-selective targeting? Why wouldn’t similar toxins evolve to target mammalian Nav channels instead?”

RESPONSE #1: We thank the reviewer for this insightful question. The original statement was indeed not fully accurate. Animal venoms are complex mixtures containing various components that target both insect and mammalian ion channels. For example, scorpion venoms typically contain a balance of insect-selective and mammal-selective α -toxins, which share a conserved $\beta\alpha\beta\beta$ scaffold but differ at key residues responsible for Na_v subtype specificity. Additionally, there are α -like toxins with dual activity against both insect and mammalian Na_v channels, though these are generally less abundant. These three classes of toxins, represented by LqhαIT, Lqh2, and Lqh3, respectively, have all been identified in the venom of *Leiurus hebraeus*. Our hypothesis is that evolutionary pressures have driven the development of both insect- and mammal-selective toxins in venomous species such as scorpions and spiders. Insects are often the predominant prey for many of these species, creating strong selection pressure for the evolution of toxins that specifically target insect Na_v channels to ensure efficient prey immobilization. Conversely, mammal-selective toxins likely serve defensive roles against vertebrate predators. Thus, insect selectivity likely reflects a broader venom evolution strategy, shaped by the ecological roles and functional demands of the venomous species. We have revised the Introduction (Line 92-94) to clarify this point: “In nature, animal venoms are complex mixtures containing multiple components that target ion channels from both insect and mammalian systems, serving functions related to prey capture and predator defense, respectively. Many animal toxins have evolved to selectively target insect Na_v channels, either by modifying channel gating or through direct channel blocking¹⁹.”

COMMENT #2: Line 147: The authors used NavPaS for structural studies, whereas BgNav1-1a was employed for electrophysiological analyses. Why not use BgNav1-1a for the cryo-EM studies as well? Doing so could provide a more direct correlation between structure and function."

RESPONSE #2: We agree that using the same Na_v subtype for both structural and functional studies would provide the most direct correlation. However, we initially attempted to express Bg Na_v for cryo-EM analysis but were unable to obtain sufficient amounts of stable, homogeneous protein for structure determination. This may be due to instability in certain intracellular loop regions or other factors affecting protein folding and expression in heterologous systems. In contrast, NavPaS has been widely used as a structural model for

insect Nav channels, and its robust expression and stability make it suitable for cryo-EM analysis. Importantly, NavPaS and BgNav1-1a share high sequence similarity, especially in the toxin-binding regions, allowing for meaningful structure-function correlations across both systems. To clarify, we have added the following text (Line 150-155): “Therefore, for electrophysiological analyses, we employed the orthologous insect Nav channel, BgNav1-1a, from the German cockroach *Blattella germanica*, which was also used in a previous study to functionally validate the NavPaS-Dc1a complex structure²⁹. BgNav1-1a shares approximately 59% sequence identity with NavPaS, with higher conservation in the extracellular toxin-interacting regions, making it a suitable surrogate for electrophysiological studies^{36, 37}.”

COMMENT #3: Line 150: Is there a specific reason the authors chose human Nav1.5 as a surrogate for other human Nav channels? It would be helpful if the authors could elaborate further—based on the resolved structures—on whether the observed selectivity toward Nav1.5 might also extend to other human Nav isoforms.

RESPONSE #3: We thank the reviewer for this important question. We selected hNav1.5 as a surrogate for mammalian Nav channels for several reasons. hNav1.5 plays a critical role in cardiac excitability and is known for its robust expression in heterologous systems, making it highly suitable for functional assays. Due to its physiological importance and sensitivity to pharmacological modulation, it serves as a conservative and practical model for evaluating potential off-target effects and mammalian safety. To further assess whether the observed toxin selectivity might extend to other human Nav isoforms, we performed a sequence alignment and structural comparison of the key toxin-binding regions, primarily VSD4 and the extracellular loops of PD1, across hNav1.1 to hNav1.9. These analyses show that residues in human Nav channels corresponding to the toxin-binding sites in NavPaS are highly conserved. Therefore, the trends observed with hNav1.5 are likely to be representative of broader mammalian Nav channel selectivity. We have added this clarification to the Results section and included a new supplementary figure (Supplementary Fig. 2C, D) showing the structural alignment of the corresponding toxin-binding regions across hNav1.1-1.8. Specifically, the revised text now reads: (Line 155-158) “As a control, we used the cardiac isoform of the human Nav channel (hNav1.5), which provides preliminary assurance of mammalian safety. hNav1.5 is representative of mammalian Nav channels and shares 42% overall sequence identity with NavPaS.”, (Line 275-278) “Sequence- and structure-based alignment of the toxin-binding regions revealed a high degree of conservation across hNav1.1-1.9, suggesting that Av3’s lack of activity against hNav1.5 likely extends to other human Nav isoforms (Fig. 3H, Supplementary Fig. 2C, D).”, and (Line 352-354) “Given the conserved nature of these binding interfaces, the observed insect selectivity of LqhαIT is also likely to apply across mammalian Nav isoforms (Fig. 4I, Supplementary Fig. 2C, D).”

COMMENT #4: Line 222: The authors state that ‘key residues at the VSD4 interface, including Pro5, Tyr7, Trp13, and Tyr18, are crucial for Av3 binding.’ However, based on Fig. 3c and the provided PDB file, Pro5 does not appear to participate in any direct interaction with the channel. Furthermore, the authors state, ‘Alanine substitutions at these residues nearly abolish binding, as confirmed by surface plasmon resonance (SPR) (Supplementary Table 2)’. This appears inconsistent with the data (Supplementary Table 2), as the P5A mutation shows a binding affinity similar to or slightly better than the wild-type (KD: 20.4 nM vs. 21.5 nM). The authors should revise this statement to accurately reflect the experimental results.”

RESPONSE #4: We thank the reviewer for this careful and constructive observation. Pro5 indeed does not form direct side-chain interactions with the channel in the complex structure. However, it makes close contact with the main-chain oxygen of Lys1256 in NavPaS, suggesting a possible structural role. As noted, the P5A mutation does not significantly alter binding affinity in the SPR assay (KD: 20.4 nM vs. 21.5 nM for wild-type). However, electrophysiological recordings reveal a ~2-fold reduction in Av3 activity, indicating a potential functional role in modulating toxin efficacy, possibly by influencing toxin orientation or stability at the interface. We have revised the original sentence in the Results section to more accurately reflect these findings (Line 234-240): “Key residues at the VSD4 interface, including Pro5, Tyr7, Trp13, and Tyr18, contribute to Av3 activity (Fig. 3C). Electrophysiological assays show that alanine substitutions at these positions reduce Av3 efficacy by 2.1-, 2.6-, 25.3-, and 7.2-fold, respectively, highlighting their functional relevance (Fig. 3F). Surface plasmon resonance (SPR) analysis further confirmed a marked loss of binding for the Y7A, W13A, and Y18A mutants (Supplementary Table 2).”

COMMENT #5: Line 274-277, 284-286: Could the authors provide a more detailed discussion of the mechanism underlying the significant conformational shifts of Glu15, Phe17, and Arg18?

RESPONSE #5: We thank the reviewer for this insightful comment. In the revised manuscript, we have reorganized and expanded the relevant paragraph to provide a more detailed discussion of the conformational shifts in Glu15, Phe17, and Arg18 upon Nav binding. These residues reside in the flexible β 1- α 1 loop of Lqh α IT and undergo a coordinated shift toward VSD4, likely driven by a combination of hydrophobic interactions and structural accommodation of the S3-S4 loop in VSD4. The observed movements are consistent with an induced-fit mechanism, facilitating tighter interface packing and enhancing toxin-channel affinity. Interestingly, Arg18, despite not forming direct contacts with NavPaS in the resolved structure, is functionally essential, the R18D mutation abolishes binding in electrophysiological assays. We speculate that Arg18 may participate in electrostatic steering or transient membrane interactions during the initial docking process, especially given the negatively charged membrane surface. We have added the following clarification to the revised text (Line 301-310): “However, the side chains of several residues near the Lqh α IT-VSD4 interface, including Glu15, Phe17, and Arg18, undergo coordinated conformational shifts toward the S3-S4 loop upon binding (Fig. 4D). These movements likely reflect an induced-fit mechanism driven by hydrophobic packing, electrostatic complementarity, and local structural accommodation. Among them, the conserved Arg18 rotates ~120°, but does not form direct contacts with NavPaS. However, the R18D mutation abolishes toxin activity, suggesting an indirect role in toxin-channel recognition, as confirmed by electrophysiology and SPR (Fig. 4F, Supplementary Table 2). We propose that Arg18 may assist in toxin orientation via electrostatic steering or transient membrane interactions prior to stable binding.”

COMMENT #6: Figure 3H: Since the authors state in the legend that invertebrate-specific residues are marked with an asterisk, it is confusing that M281 is marked this way, suggesting it is invertebrate-specific. However, based on Figure 3H, M281 appears to be conserved across both invertebrate and mammalian Nav channels, except for hNav1.8, where it is a lysine (K). This should be clarified.

RESPONSE #6: We thank the reviewer for highlighting this ambiguity and apologize for the oversight. We agree that Met281 is not strictly invertebrate-specific, as it is conserved across

most mammalian Nav channels except for hNav1.8, where it is substituted with a lysine. To avoid confusion, we have revised Figure 3H so that asterisks now indicate only residues that are strictly invertebrate-specific. This correction aligns with the sequence alignment data and ensures accurate interpretation of species-selective features.

COMMENT #7: Additionally, in the main text (lines 253–257), the authors state: ‘The selectivity is attributed primarily to invertebrate-specific residues in PD1, including Ile279, Val283, Trp391, His392, and Leu394 (Fig. 3H). Substituting these invertebrate-specific residues with their mammalian counterparts significantly reduces the efficacy of Av3 by 2.6- to 44-fold, underscoring their importance in species selectivity (Fig. 3G).’ However, Figure 3G does not include data for the Ile279 and Leu394 mutation. The authors should either provide the corresponding data or revise the text to accurately reflect the figure contents.

RESPONSE #7: We thank the reviewer for pointing out this inconsistency and apologize for the oversight. While our structural analysis identified five invertebrate-specific residues in PD1, including Ile279, Val283, Trp391, His392, and Leu394, we selected three of these residues (Val283, Trp391, and His392) located in close proximity to the toxin-binding interface for electrophysiological validation. Mutations at Ile279 and Leu394 were not functionally tested. To reflect this accurately, we have revised the main text as follows (Line 271-275): “Our complex structure identified five invertebrate-specific residues in PD1, including Ile279, Val283, Trp391, His392, and Leu394 (Fig. 3D, 3H). Electrophysiological assays demonstrated that substituting Val283, Trp391, and His392 with their mammalian counterparts significantly reduced the efficacy of Av3 by 3.0-, 16.3- and 49.4-fold, respectively (Fig. 3G), underscoring their importance in species selectivity.”

Minor points:

COMMENT #8: Line 204-206: Add reference for this statement ‘This mechanism aligns with the structural changes induced by the toxin AaH2, further supporting a common role of electrostatic interactions in toxin-induced channel modulation.’

RESPONSE #8: We have added a reference to support the statement.

COMMENT #9: Line 226, line 256, line 887, line 899: The authors should clarify which assay was used to measure Av3 activity here, in order to distinguish it from the previously mentioned SPR assay.”

RESPONSE #9: We have revised the relevant text and figure legends to clearly indicate that toxin activity was assessed using electrophysiological assays, thereby distinguishing it from the SPR-based binding measurements mentioned elsewhere.

COMMENT #10: Line 239: The authors state that ‘mutations at these residues could disrupt interactions with both the Nav channel and lipid membrane,’ which implies that this is a speculative statement. However, Figure 3F clearly demonstrates that the W8A and W13A

mutations substantially reduced activity. The authors should revise this sentence to reflect the experimental evidence rather than presenting it as a hypothesis.

RESPONSE #10: We thank the reviewer for this insightful comment. We agree that the original wording appeared speculative and have revised the sentence to more directly reflect the experimental evidence. The updated text (Line 249-259) now reads: “Additionally, some hydrophobic residues in Av3, such as Trp8 and Trp13, face the toxin-lipid interface in our complex structure, suggesting they may help stabilize the toxin conformation through interactions with the lipid membrane. A well-defined lipid-like density was observed near Trp8 of Av3 and Trp391 of PD1 (Fig. 3E). Based on its shape and position, analogous to lipids observed at equivalent sites in other human Nav structures, it was modeled as a putative phospholipid, suggesting a conserved lipid-channel interaction⁴¹⁻⁴³. Notably, Av3 does not displace the lipid but rather ‘leans’ on it, forming a sandwich-like arrangement with NavPaS. Substitution of Trp8 and Trp13 with alanine markedly reduced Av3 activity (Fig. 3F), supporting a contributing role of membrane interactions in Av3 function.”

COMMENT #11: Line 244: The reference and PDB ID for the apo structure of Av3 should be provided.

RESPONSE #11: We have added the appropriate reference for the apo Av3 structure in the main text and included its PDB ID (1ANS) in the legend of Figure 3A to enhance clarity and traceability.

COMMENT #12: Line 271: The authors should add the data source (figure or table) supporting the statement ‘... LqhαIT activity was reduced by 40-fold and 37-fold, respectively.’

RESPONSE #12: We have now cited Figure 4F in the revised text as the supporting data source for the reported reductions in LqhαIT activity, ensuring the statement is clearly referenced.

COMMENT #13: Line 304: The author should also discuss the amino acid differences in S3-S4 linker between insect and mammal sodium channels.

RESPONSE #13: We have added a brief discussion in the revised text to highlight amino acid differences in the S3-S4 linker between insect and mammalian Nav channels, and their contribution to toxin selectivity. The updated sentence reads (Line 334-337): “On the channel side, substituting the insect-specific Ile1253 in the S3-S4 linker with leucine, the corresponding residue in mammalian Navs, leads to a significant reduction in LqhαIT activity (Fig. 4G, I), highlighting the importance of species-specific interactions in determining toxin sensitivity.”

COMMENT #14 Line 309: The authors should provide reference and/or figure source to support the statement ‘While the glycans near the glycosylation site are relatively conserved across Nav channels, their distal regions vary’

RESPONSE #14: We have now cited Supplementary Fig. 4 in the revised manuscript to support the statement.

COMMENT #15: Line 585, Line 620: 50 e⁻/Å² should be 50 e⁻/Å². 3,53,627 particles should be 353,627 particles.

RESPONSE #15: We thank the reviewer for pointing out these formatting errors. We have corrected "50 e⁻/Å²" to "50 e⁻/Å²" and "3,53,627" to "353,627" in the revised manuscript.

COMMENT #16: Line 711: The journal name Annual Review of Entomology is not in the abbreviated format. Please review all references to ensure consistent formatting throughout the reference list.

RESPONSE #16: We thank the reviewer for noting this inconsistency. The journal name Annual Review of Entomology has been updated to its abbreviated format, and we have reviewed all references to ensure consistent formatting.

COMMENT #17: Line 920-924: The authors showed EeNav1.4 in supplementary Fig.4 but did not mention it in legend.

RESPONSE #17: We have updated the legend of Supplementary Fig. 4 and included "EeNav_v1.4 (PDB ID: 5XSY)" for clarity and completeness.

COMMENT #18: Line 950-951: The authors should clarify the meaning of 'N.D.' to avoid ambiguity.

RESPONSE #18: To avoid ambiguity, we have updated the figure legend to clarify that "N.D." stands for "KD not detectable".

COMMENT #19: Figure 1A: To provide a more complete picture of the electrophysiological effects, the authors should show representative current traces of Av3 and LqhαIT on hNav1.5.

RESPONSE #19: We thank the reviewer for this helpful suggestion. We have now included representative current traces of both Av3 and LqhαIT on hNav1.5 in the revised Fig. 1A. These traces demonstrate the minimal or undetectable effects of either toxin on hNav1.5 at the same concentrations used for BgNav_v, consistent with their insect-selective activity profiles.

COMMENT #20: Figure 1B: The hNav1.5 curve for Av3 appears relatively smooth, whereas the curve for LqhαIT shows large error bars. The authors should provide an explanation for this discrepancy. In addition, the figure legend should clearly indicate whether the data are presented as mean ± SD or mean ± SEM, and the number of replicates (n) should be specified."

RESPONSE #20: We thank the reviewer for pointing out this discrepancy. The large error bars observed for the LqhαIT curve on hNav_v1.5 were attributable to the lower quality of the oocytes used in that particular batch. To address this, we repeated the LqhαIT experiments on hNav_v1.5 using stricter oocyte selection criteria. The updated results now show error levels comparable to those of the Av3 group. Figure 1B has been revised accordingly. In addition, we have clarified in the figure legend that data are presented as mean ± SEM and have specified the number of replicates (n = 3).

COMMENT #21: Figure 2B: The authors should indicate what the dashed and solid lines represent in the right panel for clarity.

RESPONSE #21: We have updated the Fig. 2B legend to clarify that: "Several salt bridge pairings are altered upon toxin binding. Solid lines indicate salt bridges present in the current state, while dashed lines denote those formed in an alternate state."

COMMENT #22: Figure 3E: The authors should clarify how the lipid was identified as a phospholipid based on the density map. Additionally, the sigma value used for contouring the map should be specified in the figure legend.

RESPONSE #22: We thank the reviewer for the suggestion. The lipid was modeled as a putative phospholipid based on its well-defined density and by analogy to previous cryo-EM structures of human Na_v channels—including hNa_v1.3, hNa_v1.7, and hNa_v1.8 (PDB IDs: 7W77, 8F0Q, and 9DBK)—where similar lipid densities were modeled at equivalent membrane-facing sites. This comparison supports the plausibility of conserved lipid binding in our structure. We have revised the main text to describe the lipid as a putative phospholipid and updated the figure legend accordingly. The density is contoured at 5 σ , as indicated in the revised legend. The updated text in the Results section reads (Line 251-255): "A well-defined lipid-like density was observed near Trp8 of Av3 and Trp391 of PD1 (Fig. 3E). Based on its shape and position, analogous to lipids observed at equivalent sites in other human Na_v structures, it was modeled as a putative phospholipid, suggesting a conserved lipid-channel interaction⁴¹⁻⁴³." The revised figure legend now reads: "A putative phospholipid was modeled into a well-defined density (contoured at 5 σ) located between Trp8 of Av3 and Trp391 of PD1, suggesting a potential role in mediating the interaction between Av3 and Na_vPaS."

COMMENT #23: Figure 3F, Figure 3G, Figure 4F, Figure 4G, Figure 5B, Figure 5C, Figure 5D: To enhance transparency and allow better interpretation of data variability, the authors are encouraged to display individual replicate values and specify the number of replicates (n) in the bar plots.

RESPONSE #23: We thank the reviewer for this valuable suggestion. In the revised figures, we have updated the bar plots to include individual replicate values as overlaid scatter points, thereby enhancing transparency and allowing clearer visualization of data variability. The number of replicates (n = 3) is now explicitly stated in the corresponding figure legends. For the datasets in Figures 3F, 3G, 4F, and 4G, the original EC₅₀ values were derived from globally fitted dose–response curves using pooled data from three replicates, which did not permit meaningful display of inter-replicate variability. In response to the reviewer's comment, we have reanalyzed the original data by independently fitting each replicate using the Hill function and now report the mean EC₅₀ \pm SEM. These values show only minor deviations from the original results, and the overall trends and conclusions remain unchanged. For Figure 5B, the SPR measurements were performed once per mutant at a minimum of five concentrations as preliminary high-throughput screen to prioritize candidates for follow-up validation. Because no replicate experiments were conducted for this screen, individual replicate points are not available. However, the updated electrophysiological and *in vivo* assay results shown in Figures 5C and 5D, now displayed with overlaid scatter points, confirm the functional relevance of the selected mutants.

COMMENT #24: Figure 3H: D1 or DI? Please be consistent.

RESPONSE #24: We have revised the label in Figure 3H to consistently use “DI” (Domain I) throughout the figure and text for clarity and uniformity.

COMMENT #25: Figure S3A and S3D: The text ‘86 residues’ appears to be unintended and should be removed.

RESPONSE #25: The unintended text “86 residues” has been removed from Supplementary Fig. 3A and 3D in the revised version.

COMMENT #26: Table-S2: The authors also performed SPR analysis for LqhaIT variants, but this was not mentioned in the main text. Is there a specific reason for this omission?

RESPONSE #26: We thank the reviewer for this helpful comment. SPR measurements for LqhaIT variants were indeed performed and are reported in Supplementary Table S2. In the original submission, we focused on electrophysiological results in the main text to maintain clarity, as the number of LqhaIT mutants was relatively large and presenting both datasets in detail risked overcomplicating the section. To address this, we have now briefly referenced the SPR data in the main text alongside the electrophysiological results, noting the consistency between the two assays (Line 350-351): “For selected mutants, binding affinities were also assessed by SPR (Supplementary Table 2), which showed consistent trends with the electrophysiological results.”

Reviewer #3 (Remarks to the Author):

The structural basis for the action of peptide toxins acting selectively on insect sodium channels is poorly understood but can provide valuable information for the future design of effective and selective insecticides. In this study, Jiang and colleagues determined cryo-EM structures of the cockroach sodium channel NavPs in complex with either the LqhaIT or Av3 insecticidal toxin. The binding interfaces between toxin and channel were probed using mutagenesis, which included introducing mammalian substitutions into the insect channel. Mutants were tested using surface plasmon resonance studies with NavPs or two-electrode voltage-clamp studies with the BgNav1.1 channel from a different cockroach species (as NavPs is non-functional). This study also describes an AI modelling approach used to identify mutations of LqhaIT that were predicted to increase its binding affinity. One such mutant (A39L) was shown to be twice as effective as the wildtype toxin in a bioassay.

This is very interesting work that would be of great interest to ion channel pharmacologists, particularly those seeking to develop novel insecticides. One exciting finding is how different the Av3 binding site is from that of LqhaIT (as the sites have very little overlap) and how Av3 is buried at an interface between the VSD4 and pore domain. Overall, the cryo-EM work is well described and has clear, attractive figures and the electrophysiology work also clear and

straight-forward. There are a number of issues, though, that the authors should address, particularly regarding the surface plasmon resonance (SPR) work.

COMMENT #1: Lines 224-225 describe how alanine mutations of Av3 'nearly abolish binding' but that some of these alanine mutations produce only a ~2 fold reduction in Av3 activity when tested using electrophysiology. Leaving aside the issue that two different cockroach channels were used (NavPs vs BgNav1.1), it seems that the NavPs immobilisation and subsequent toxin binding steps of SPR were performed with buffers lacking detergent. This membrane protein is almost certainly to have undergone some unfolding in the absence of detergent and, although some part of the receptor site may remain to allow toxin binding, the authors cannot have confidence that NavPs is adopting a native conformation and that the results are therefore physiologically relevant. This is also an issue for the AI approach to mutant generation, as SPR was the experimental technique used to first screen AI-suggested mutations of LqhaIT (Fig 5B). I suspect the use of SPR limited the effectiveness of this approach and that valid affinity-increasing mutants were overlooked. Can the authors comment on this and, if detergent was in fact included in the buffers, please provide example sensorgrams?

RESPONSE #1: We appreciate the reviewer's thoughtful comments regarding the SPR assay conditions and their implications for both structural relevance and mutant screening. We agree that working with membrane proteins in SPR presents a challenge. This presents a dilemma: including detergent helps maintain membrane protein stability, but it can disrupt hydrophobic interactions between the toxin and the channel. Conversely, removing detergent preserves these hydrophobic interactions, but may compromise the structural integrity of the channel. To address this, we adopted a two-phase strategy: 1) During immobilization, NavPaS was maintained in a mild detergent-containing buffer (0.04% GDN) to preserve a stable, native-like conformation. 2) Following immobilization, we exchanged to detergent-free buffer for the measurement phase, and initiated the binding assay promptly to minimize potential destabilization. Given the affinity of detergents like GDN for transmembrane domains, we expect the detergent to remain associated with NavPaS for a limited duration, helping to sustain structural integrity during measurement. To validate this approach, we included wild-type toxin controls at both the beginning and end of each SPR run. The consistency in measured KD values across these controls supported the stability of NavPaS during the assay window. While we acknowledge that the absolute affinity values may be influenced by these conditions, all toxin variants were tested under identical buffer conditions, enabling meaningful relative affinity comparisons. Regarding the AI-based mutation screening, we agree that the limitations inherent to the SPR setup may have reduced the dynamic range for detecting subtle affinity improvements. However, the binding trends observed via SPR largely correlate with the follow-up electrophysiological validation data, reinforcing the utility of SPR as a first-pass screening tool to identify impactful mutations.

To clarify the experimental setup, we have updated the following description in the Methods section: "NavPaS protein was diluted to 20 µg/mL in immobilization buffer (10 mM sodium acetate, pH 4.5, 0.04% GDN) and immobilized to 15,000 RUs on a CM5 sensor chip." We compared sensorgrams obtained under both conditions: in the presence and absence of detergent (see below). While measurements with detergent showed no detectable binding, those without detergent produced well-fitted curves with clear high-affinity binding.

Figure Legend: SPR sensorgrams of LqhαIT binding to NavPaS under detergent-free (A, B) and detergent-containing (C, D) conditions.

Panel A and B present sensorgrams from two independent runs acquired in detergent-free buffer with LqhαIT concentrations from 6.25 to 100 nM at the beginning and the end of a screening experiment. Colored lines represent experimental data, and black lines show the best fits to a 1:1 binding model. Consistent KD values (83.1 and 86.5 nM) and low χ^2 values (1.03 and 1.44 RU², both < 10% Rmax) confirm the stability and reproducibility of the measurements, as well as the high quality of the model fits.

Panel C and D present sensorgrams acquired in the presence of 0.04% GDN with 6.25-400 nM LqhαIT (C) and 2.25-36 μM (D). No detectable binding was observed in both experiments under detergent-containing conditions, even with high concentrations of toxin.

COMMENT #2: Lines 195-206 describe how toxin-induced conformational changes generate binding contacts between different sections of the channel: the DIII-DIV linker, DIV S6 helix and C-terminal domain. The issue is that the amino acids involved in salt bridges & hydrogen bonds differ between NavPs and the BgNav1.1 & Nav1.5 channels: NavPs R1293 is alanine in BgNav1.1 and methionine in Nav1.5; NavPs R1138 is lysine in BgNav1.1 and asparagine in Nav1.5; NavPs D1420 is glutamate in Nav1.5. It would be helpful to show a sequence alignment of these channel sections in Fig 2 and for the authors to point out that interactions shown for the NavPs structure will differ in other channels.

RESPONSE #2: We thank the reviewer for this valuable suggestion. We agree that some of the electrostatic interactions observed in the NavPaS structure are not conserved across insect or mammalian Nav isoforms. To address this, we have added a sequence alignment in the revised Fig. 2C, highlighting key differences at the toxin-induced electrostatic interfaces, including residues involved in salt bridges. We have also revised the main text to clarify that the several of the described interactions are specific to the NavPaS structure, and that the exact interaction network may differ in other Nav channels due to sequence variation. The updated Results section now reads (Line 214-218): “However, several key interacting residues in NavPaS, such as Arg1138 and Arg1293, are substituted by neutral residues (e.g., alanine, methionine, or

asparagine) in other Nav isoforms (Fig. 2C). These differences suggest that the precise electrostatic network and conformational changes induced by toxin binding may not be fully conserved across other Nav channels.”

COMMENT #3: Lines 257-259 speculate that “variations in glycosylation patterns may influence toxin selectivity” and, in lines 309-312, that the distal regions of glycans vary. While this is an intriguing suggestion, the authors should provide a reference(s) that describes differences in glycosylation between insects and mammals. There is also the issue that LqhαIT mutants with substitutions from the mammalian-selective AaH2 toxin were tested electrophysiologically in amphibian (*Xenopus*) cells and so the claim based on results shown in Fig 4F that interactions between LqhαIT and the glycan “highlights the critical role of this region in determining selectivity” is highly speculative.

RESPONSE #3: We thank the reviewer for this thoughtful and important comment. We fully agree that the native glycosylation patterns of insect and mammalian Nav channels remain poorly defined, and that current structural data provide only partial insights due to the use of heterologous expression systems. Cryo-EM structures of several Nav channels, including NavPaS, hNav1.1-1.6, and hNav1.8, all expressed in HEK293 cells, consistently show glycosylation at the equivalent site to Asn330 of NavPaS. However, the observed glycan structures differ, suggesting that both the local 3D context and the expression system (e.g., HEK293 vs. *Xenopus* oocytes) influence glycan composition and presentation. We acknowledge that the glycosylation in *Xenopus* oocytes, used in our electrophysiological assays, may not accurately reflect the native patterns of either insect or mammalian Nav channels. Thus, while the data suggest a potential contribution of the glycan region to toxin selectivity, we agree that the conclusion should be more cautiously framed. In response, we have revised the text to avoid overinterpretation and now refer to the glycan as a plausible contributor, rather than asserting a critical role. We have also cited Supplementary Fig. 4, which compares glycan presentation across multiple Nav structures, further supporting this hypothesis as a direction for future study. The revised text in the Results section now reads (Line 341-348): “While the glycans near the conserved glycosylation site are relatively similar across Nav channels, their distal regions vary (Supplementary Fig. 4). Substituting insect-selective LqhαIT sequences in this region with those from mammal-selective AaH2 greatly reduces activity, including the N9D, Y10V, I57T, and V59G substitutions which significantly decrease LqhαIT activity (Fig. 4B, H, F). Although glycosylation in *Xenopus* oocytes may not reflect the native insect or mammalian patterns, the observed differences suggest that the distal glycan structure may plausibly contribute to differential toxin binding and selectivity.”

COMMENT #4: ‘N=’ values should be reported throughout e.g. in figures with electrophysiology and bioassay data.

RESPONSE #4: We have carefully reviewed all figures containing electrophysiology and bioassay data and have added the number of replicates (n = 3) across the figure panels and legends to ensure clear and transparent reporting throughout the manuscript.

MINOR POINTS:

COMMENT #5: Lines 41-43: It is not clear why the Av3-bound channel is proposed to be in a 'deactivated state' while the LqhαIT-bound channel has 'disrupted fast inactivation'. The ion-conducting pores of apo-, Av3- and LqhαIT structures look identical (sup Fig 3C, F) and both toxins inhibited fast inactivation. It doesn't make sense to distinguish the two toxins as one caused deactivation and the other causing disrupted fast inactivation.

RESPONSE #5: We thank the reviewer for pointing this out and agree that the original wording may have led to confusion. While Av3 and LqhαIT engage distinct binding sites on NavPaS, both toxins ultimately exert a similar functional effect, disruption of fast inactivation, by stabilizing VSD4 in a deactivated conformation. To clarify this, we have revised the abstract to emphasize the shared mechanism while still noting the distinct binding modes of the two toxins. The revised text now reads (Line 38-42): "Both toxins bind to the voltage-sensing domain 4 (VSD4) of NavPaS and disrupt fast inactivation by stabilizing the S4 segment in a deactivated conformation. While Av3 engages a previously uncharacterized membrane-embedded site between VSD4 and pore domain 1 (PD1), LqhαIT binds to the classical neurotoxin site 3, illustrating distinct binding modes that converge on a shared mechanism of action."

COMMENT #6: L47: "remarkable doubling in efficacy". It might help to point out that this was tested by bioassay

RESPONSE #6: We have clarified in the revised manuscript that the observed increase in efficacy was measured through insecticidal bioassays. The updated sentence now reads (Line 45-47): "Leveraging these insights, we applied AI-driven protein design tools to increase the insecticidal potency of LqhαIT, resulting in a variant with a remarkable doubling in efficacy, as confirmed by insecticidal bioassays."

COMMENT #7: L71: "calmoduzlin" ... calmodulin

RESPONSE #7: The misspelling of "calmodulin" has been corrected in the revised manuscript.

COMMENT #8: L201: "Arg1277-Glu1435". It might be better to change Arg1277 to 'R6' to be consistent with the figure. Alternatively, "Arg1277 (R6)-Glu1435" could work

RESPONSE #8: To ensure consistency with the figure while retaining clarity in the text, we have revised the phrasing to "Arg1277 (R6)-Glu1435."

COMMENT #9: L202: "binding results in only the first two bridges" ... this is not specific enough, especially as three bridges are shown in Fig 2B

RESPONSE #9: We thank the reviewer for pointing this out. To ensure consistency with Fig. 2B and improve clarity, we have revised the sentence in the manuscript to be more specific. The updated text now reads (Line 206-210): "Av3 induces the formation of three additional electrostatic bridges, including Arg1277 (R6)-Glu1435, Arg1293-Asp1427, and Asp1420-Arg1142. In contrast, LqhαIT binding induces only two of these bridges, including Arg1277 (R6)-Glu1435 and Asp1420-Arg1142, alongside the pre-existing Asp1420-Arg1138 interaction (Fig. 2B)."

COMMENT #10: L202: “Pro5, Tyr7, Trp13, and Tyr18, are crucial for Av3 binding (Fig. 3C). Alanine substitutions at these residues nearly abolish binding, as confirmed by surface plasmon resonance (SPR)”. The SPR results for Y7, W13 and Y18 are “N.D.”, which stand for ‘not determined’, indicating no data was collected on them. In addition, the SPR data shows the P5 mutant has a similar affinity as wildtype. These two sentences therefore need clarification.

RESPONSE #10: We thank the reviewer for this important clarification and apologize for the misleading wording in the original text. The SPR measurements for the Y7A, W13A, and Y18A mutants were performed, but the binding signals were below the detection threshold under the experimental conditions, and thus no KD values could be determined. To avoid confusion with uncollected or missing data, we have updated the figure legend of Supplementary Table 2 to clarify that “N.D.” refers to “KD not detectable.” Regarding the P5A mutant, we agree that its binding affinity is comparable to wildtype based on the SPR data. Accordingly, we have revised the main text to remove Pro5 from the list of residues associated with reduced binding, ensuring full consistency with the experimental results.

COMMENT #11: L230: “reduce the binding affinity and efficacy of Av3, with Trp8 ... having more pronounced effect”. SPR data shows ‘N.D.’ for W8 and so binding affinity wasn’t determined.

RESPONSE #11: As noted in the last response, the W8A mutant was tested by SPR, but the binding signal was below the detection threshold, and therefore no KD value could be determined. The label “N.D.” in Supplementary Table 2 denotes “KD not detectable”, indicating that the measurement was performed but yielded no quantifiable binding under the tested conditions, not that the experiment was omitted.

COMMENT #12: L280: “the D1252R mutation completely abolishes it (Fig 4C, G)”. The D1252R mutation is not shown in Fig 4G.

RESPONSE #12: We thank the reviewer for pointing out this oversight. The D1252R mutant was tested in electrophysiological recordings, and no detectable response to LqhαIT was observed under the experimental conditions. To accurately reflect this result, we have now added a gray bar labeled “N.D.” (activity not detectable) to Figure 4G, and the figure legend has been updated accordingly. This revision ensures that the statement “completely abolishes it” is appropriately supported by the data presented in the figure.

COMMENT #13: L311: “Fig. 4E, H, F” ... I believe this should cite Fig 4B instead of Fig 4E.

RESPONSE #13: We have corrected this in the revised version.

COMMENT #14: L348: “we applied manual filtering based on empirical knowledge”. This requires further explanation i.e. what empirical knowledge were you acting on?

RESPONSE #14: We thank the reviewer for requesting clarification. The phrase “manual filtering based on empirical knowledge” has now been revised for greater transparency and consistency. As detailed in our response to Comment #6 of Reviewer 1, this filtering was not based on subjective judgment but rather on reproducible structural and biophysical criteria.

Specifically, we prioritized mutations that: (i) were located at the toxin-channel interface, as defined by our cryo-EM structure; (ii) were predicted to improve hydrophobic packing or electrostatic interactions; and (iii) did not involve residues critical for maintaining structural integrity, such as those forming disulfide bonds or key secondary structure elements. These criteria are now clearly described in the Methods section to ensure reproducibility.

COMMENT #15: L414: “The distal region of the glycosylation site at Asn330 has diverged during evolution (Supplementary Fig. 4)”. Sup Fig 4 shows the glycan from a number of sodium channel cryo-EM structures. However, all of these channels (or the majority, including NavPs) were expressed in the mammalian HEK cell system and so Sup Fig 4 cannot be used to support this statement about glycosylation differences during evolution.

RESPONSE #15: We thank the reviewer for pointing this out. We agree that Supplementary Fig. 4, which presents glycan densities from Nav channels primarily expressed in mammalian HEK cells, does not provide sufficient evidence to support a definitive conclusion about evolutionary divergence in glycosylation. The observed variation in the distal conformation of the glycan at Asn330 likely reflects a combination of factors, including expression system-dependent effects and intrinsic sequence- or structure-related differences that may have evolved among Nav isoforms. However, distinguishing between these contributions remains challenging. Regardless of their origin, these structural differences in glycan presentation may still alter the geometry of the toxin-glycan interface and contribute to species-selective toxin recognition. To clarify this point, we have revised the Discussion to avoid strong evolutionary interpretations and focus instead on structural heterogeneity. The previous statement has been replaced with (Line 456-464): “Additionally, while the basal portion of the glycosylation at Asn330 is relatively conserved across Nav channels, the distal region displays structural heterogeneity among different Nav isoforms (Supplementary Fig. 4). Notably, the CTD and the β 1- α 1 loop of Lqh α IT, which interact with this distal region, are also conserved among insect-selective α -ScTxs, suggesting a potential role for this glycan surface in modulating toxin-channel recognition (Fig. 4H). These structural features in insect Navs present opportunities for the future design of biopesticides. Exploiting these distinctions could further enhance the insect selectivity of peptide toxins, minimizing off-target effects and increasing their effectiveness.”

COMMENT #16: L860: “R3 crossing the hydrophobic constriction site”. Please check as I believe it is R4 that crosses the HCS.

RESPONSE #16: We thank the reviewer for catching this error. We have corrected the text in the revised manuscript to indicate that it is R4, not R3, that crosses the hydrophobic constriction site (HCS).

COMMENT #17: L869: “superimposed onto with its apo state” ... no need for the word ‘with’

RESPONSE #17: The phrasing has been corrected in the revised manuscript by removing the unnecessary word “with.”

COMMENT #18: L877: “invertebrate-specific residues are marked with an asterisk”. Some residues are inappropriately marked with an asterisk as they are conserved between insect and

most/all-but-one mammalian channels. Please be more selective in the use of asterisks to mark invertebrate-specific residues.

RESPONSE #18: We have carefully re-examined the sequence alignments and updated the asterisk markings in Fig. 3 to indicate only strictly invertebrate-specific residues. Residues that are conserved in most or all mammalian Na_v channels have been unmarked to ensure accurate and selective annotation.

COMMENT #19: L892: “invertebrate-specific with an asterisk” ... please be more selective in the use of asterisks to mark invertebrate-specific residues.

RESPONSE #19: As noted in Response #18, we have revised the asterisk markings in Fig. 4 to more selectively indicate only strictly invertebrate-specific residues, ensuring accurate representation of sequence conservation.

COMMENT #20: L929: “Bar chart ranking prediction models, showing the superior accuracy of ComplexDDG”. There is insufficient detail here or in Methods or elsewhere in the text describing how this comparison between different softwares was performed.

RESPONSE #20: We thank the reviewer for highlighting this omission. To address this, we have expanded the Methods section to provide a detailed description of the model benchmarking process as follows (Line 701-708): “To evaluate the predictive accuracy of ComplexDDG, we benchmarked the model using the s1131 dataset from the SKEMPI database, which comprises 1,131 non-redundant single-point mutations across 144 protein-protein complexes^{48, 49}. Mutant structures were generated using FoldX based on the wild-type crystal structures, and incompatible PDB files were preprocessed with PDBFixer v1.8. A fivefold split-by-complex cross-validation strategy was applied, with ~20% of complexes held out per fold to prevent data leakage. ComplexDDG’s performance was compared against representative methods including GeoPPI, DDGPredictor, TopGBT, and FoldX⁵⁰⁻⁵³.”

COMMENT #21: Sup Fig 3C, F. The side chains shown in the pore should be identified and the reason for showing them stated.

RESPONSE #21: In Supplementary Fig. 3C and 3F, the side chains displayed as sticks represent key hydrophobic residues at the activation gate that define the intracellular pore constriction. We have now labeled these residues in the figure and revised the legend accordingly: “Top view of the activation gate, highlighting two layers of hydrophobic residues (shown as sticks) that define the pore constriction, indicating an unaltered pore conformation across the structures.”

Best regards,

Zhiguang Yuchi

Dear Reviewers,

We sincerely thank the reviewers for their insightful feedback and constructive suggestions that have substantially improved our work. Below, we provide a point-by-point response to their comments. The reviewers' comments are reproduced in dark red, followed by our responses in dark blue.

Reviewer #2 (Remarks to the Author):

COMMENT: The authors have adequately addressed all of my previous concerns. I recommend acceptance of the manuscript in its current form.

RESPONSE: We thank the reviewer for the positive evaluation and recommendation for acceptance.

Reviewer #3 (Remarks to the Author):

COMMENT: The authors have addressed all my concerns and their changes have strengthened the manuscript.

One of my major concerns was about the SPR studies carried out in the absence of detergent. In their reply to reviewers, the authors have made a clear case for why they performed their SPR studies without detergent and have supplied a convincing figure with sensograms showing (a) that detergent interferes with toxin binding and (b) that binding curves of wildtype LqhαIT "at the beginning and the end of a screening experiment" look almost identical. My confidence in the SPR data has consequently greatly increased. I believe it would be beneficial for readers (and especially those planning to replicate this approach with their own toxin/binder of interest), that this figure is added as a supplementary figure to the manuscript. A sentence or two should be added to the Results about the negative effect of detergent on ligand binding and that control toxin bindings were carried at the beginning and end of a screening experiment to verify integrity of the receptor site. If the immobilised channel was washed with buffer containing detergent in between toxin binding runs then this would be an important point to also add.

RESPONSE: We thank the reviewer for the positive and constructive feedback. We confirm that the immobilized channel was not washed with detergent-containing buffer between runs. Following the reviewer's suggestion, we have added a new supplementary figure demonstrating that the presence of detergent markedly reduced toxin-channel binding, and that wild-type LqhαIT controls, when injected at the beginning and end of each screening experiment, produced highly consistent sensograms, confirming the stability of the immobilized channel. To maintain the flow of the Results section, a corresponding clarification has been added to the Methods section. The revised text now reads: "As the inclusion of detergent impaired toxin-channel binding (Supplementary Fig. 5A), all measurements were performed in detergent-free buffer. The stability of the immobilized channel was confirmed by injecting wild-type LqhαIT at the beginning and end of each screening experiment, which produced highly consistent response curves (Supplementary Fig. 5B, C)."

Reviewer #4 (Remarks to the Author):

The discussion of their self-developed “ComplexDDG” algorithm is not convincing and may need to be improved.

COMMENT #1: There is no detailed (or even a general) discussion of what are the main idea, method, architecture, or innovation of the newly proposed “ComplexDDG” model. It is totally unclear the performance of this model is due to model innovation, pretrain knowledge, or even just some leaking of the data. Without a good discussion of the details, it is impossible to properly evaluate the performance of the model.

RESPONSE #1: We thank the reviewer for this insightful comment and for highlighting the need for a clearer explanation of our self-developed ComplexDDG algorithm. We fully agree that transparent descriptions of the model's architecture, methodology, and innovations are essential for evaluating its performance and novelty. To address this concern, we have added a new Supplementary Note that provides detailed information on the model architecture, performance evaluation, and ablation studies, along with corresponding figures and tables.

In brief, ComplexDDG is a deep regression model designed to predict changes in binding free energy ($\Delta\Delta G$) upon mutation by integrating sequence-, structure-, and energy-based information. The model takes two major types of inputs: 1. Vector representations that fuse global evolutionary and perturbation features extracted using the pretrained ESM-1v language model and Biopython, respectively; 2. Twenty-two energy terms computed by FoldX. The key innovation of ComplexDDG lies in the direct incorporation of high-order sequence embeddings from ESM-1v as node features within a geometric attention-based network. By jointly embedding these representations with the energy-derived features, the model learns a comprehensive latent representation that effectively captures the mutational impacts on binding affinity. The final joint embedding is passed to a regression layer that outputs $\Delta\Delta G$ predictions with both high accuracy and interpretability (Supplementary Fig. 9). We have also included benchmark comparisons against existing $\Delta\Delta G$ predictors, and ablation studies to assess the contribution of each feature component. Our results show that ComplexDDG achieves significantly improved PCC and RMSE values compared to DDGPredictor, Rosetta, FoldX, and GeoPPI, and performs comparably to the latest GearBind model with higher robustness. In ablation experiments, removal of FoldX and ESM-1v features reduced PCC by 0.173 and 0.064, respectively. Importantly, all evaluations were conducted using a split-by-complex, fivefold cross-validation, where data from the same complex were confined to a single fold, ensuring that the observed performance improvements are not attributable to data leakage.

Due to commercial confidentiality, we are unable to release the full source code at this stage, but we plan to make it publicly available in the future. We hope that the newly added Supplementary Note provides sufficient transparency to clarify the design rationale, methodological innovations, and strengths of the ComplexDDG model.

COMMENT #2: There are various mistakes in the comparison with state-of-the-art models. For instance, the authors compare their model with “TopNetTree” and “TopGBT” models and they cite the reference 51, McMurray HR, et al. Gene network modeling via TopNet reveals functional dependencies between diverse tumor-critical mediator genes. Cell Rep 37, 110136 (2021). It is obvious that this is the wrong reference. The “TopNet” model in ref 51 IS NOT the “TopNetTree” and “TopGBT” models for protein-protein binding affinity change upon mutation. In fact, the right reference should be Mengjun Wang, Z. X. Cang, and Guo-Wei Wei, A topology-based network

tree for the prediction of protein-protein binding free energy changes following mutation, *Nature Machine Intelligence*, 2, 116-123 (2020).

RESPONSE #2: We thank the reviewer for carefully identifying this miscitation and fully agree with the concern. We sincerely apologize for the earlier error. The correct reference for the topology-based methods is indeed Wang, Cang & Wei, *Nature Machine Intelligence*, 2, 116-123 (2020), which describes the TopNetTree/TopGBT models for predicting protein-protein binding free energy changes upon mutation.

During revision, we initially attempted to include TopNetTree and TopGBT in our benchmarking alongside other state-of-the-art $\Delta\Delta G$ predictors. However, the available resources appeared to contain incomplete training files and configuration information, and our preliminary runs produced atypically low and inconsistent scores. To ensure a fair and reproducible comparison, we therefore replaced these models with the GearBind predictor, a recent and well-documented mainstream baseline. All retained baseline models, including GearBind, GeoPPI, Rosetta, DDGPredictor, and FoldX, were re-evaluated under identical data splits, preprocessing procedures, and evaluation metrics. These revisions improve the fairness, consistency, and clarity of our performance comparisons. We are grateful to the reviewer for highlighting this issue, which enabled us to correct the citation and further strengthen the methodological rigor of our benchmarking.

COMMENT #3: It seems that the comparison results in “Supplementary Fig. 5 Performance of the ComplexDDG model” are adopted from supplementary files of reference 53 Shan S, et al. Deep learning guided optimization of human antibody against SARS-CoV-2 878 variants with broad neutralization. *Proc Natl Acad Sci U S A* 119, e2122954119 (2022). In which, the results of “TopNetTree” and “TopGBT” models are significantly lower than all existing models. In fact, it is totally unclear how the authors in PNAS paper get the results, considering that they also cite the wrong reference in their table (Note that they cite ref 46 in their Table, but ref 46 has nothing to do with “TopNetTree” and “TopGBT” models).

RESPONSE #3: We thank the reviewer for carefully identifying this issue and sincerely apologize for the confusion. We think that the previously reported low scores for TopNetTree and TopGBT (both in our earlier version and in the cited PNAS paper) may have been influenced by the incomplete implementation resources, rather than by the inherent model performance, as noted above. To resolve this, we have reimplemented all baseline models under a unified and transparent evaluation protocol on the SKEMPI v2.0 (s1131) dataset using fivefold split-by-complex cross-validation, ensuring fair, reproducible, and internally consistent comparisons. Additionally, we now report both Pearson correlation coefficient (PCC) and root mean square error (RMSE) metrics to provide a more comprehensive assessment of predictive accuracy and robustness.

Under this standardized protocol, ComplexDDG achieved superior performance (PCC: 0.814, RMSE: 1.421) compared to other established predictors, including DDGPredictor (PCC: 0.615, RMSE: 2.082), Rosetta (PCC: 0.340, RMSE: 3.514), FoldX (PCC: 0.395, RMSE: 2.612), and GeoPPI (PCC: 0.591, RMSE: 2.034). The recently developed GearBind predictor (PCC: 0.808, RMSE: 1.473) showed comparable accuracy, but with higher standard deviation values, indicating lower robustness relative to ComplexDDG. Since the GeoPPI training code was not publicly available, we reproduced it based on the methodological descriptions from the original publication, mimicking the rotamer sampling strategy and hyperparameters selection for the gradient boosting tree (GBT) model accordingly. We have updated Supplementary Figures and

Tables to reflect these corrected results and revised the Supplementary Note to clarify the data sources and evaluation procedures.

Best regards,
Zhiguang Yuchi

Dear Reviewers,

We thank all the reviewers for the positive comment.

Reviewer #3 (Remarks to the Author):

COMMENT: The authors have made the requested changes to the manuscript and I recommend its publication.

RESPONSE: We thank the reviewer for the positive evaluation and recommendation for acceptance.

Reviewer #4 (Remarks to the Author):

COMMENT #1: All my concerns are well addressed! I have no further comments and would recommend the publication of the paper!

RESPONSE #1: We thank the reviewer for the positive evaluation and recommendation for acceptance.

Best regards,

Zhiguang Yuchi